# Capturing individual variation in children's electroencephalograms during nREM sleep

Verna Heikkinen[1,2]*, Susanne Merz[1], Riitta Salmelin[1], Sampsa Vanhatalo[3],
Leena Lauronen[3], Mia Liljeström[1,2]©, Hanna Renvall[1,2]©

**1** Department of Neuroscience and Biomedical Engineering, Aalto University School of Science, Espoo,
Finland, **2** BioMag Laboratory, HUS Medical Imaging Center, Helsinki University Hospital, Helsinki
University and Aalto University School of Science, HUS Helsinki, Finland, **3** Departments of Physiology
and Clinical Neurophysiology, BABA Center, Paediatric Research Center, Children's Hospital, Helsinki
University Hospital and University of Helsinki, Helsinki, Finland

© These authors contributed equally to this work.
* verna.heikkinen@aalto.fi

## Abstract

Human brain dynamics are highly unique between individuals: functional neuroimaging studies have recently described functional features that can be used as neural fingerprints. However, the stability of these fingerprints is affected by aging and disease. As such, the stability of brain fingerprints may be a useful metric when studying normal and pathological neurodevelopment. Before examining clinically relevant deviations, the individual stability and variation of neuroimaging features across brain maturation in normally developing children need to be addressed with real clinical data. Here we applied Bayesian reduced-rank regression (BRRR) to extract low-dimensional representations of electroencephalography (EEG) power spectra measured during different non-REM sleep stages (N1 and N2) from 782 normally developing children aged between 6 weeks to 19 years. The representations learned within specific sleep stages successfully separated between subjects and generalized across sleep stages. Fingerprint stability increased with the age of the subjects. Compared to correlation-based fingerprinting methods, the BRRR model performed better, especially in fingerprinting across sleep stages, highlighting the usefulness of dimensionality reduction when the noise and signal of interest are correlated. While further studies are needed to address the possible non-linear maturation effects over developmental periods, our results demonstrate the existence of stable within-session neurofunctional fingerprints in pediatric populations.

## Author summary

Leaning to the intuition of the uniqueness of individual brain dynamics, cortical fingerprinting based on functional brain-imaging features has gained momentum. Most fingerprinting studies have been performed on adults and not on clinical data

journal.pcbi.1013931

**Peer Review History:** PLOS recognizes
the benefits of transparency in the peer
review process; therefore, we enable the
publication of all of the content of peer
review and author responses alongside
final, published articles. The editorial
history of this article is available here:
https://doi.org/10.1371/journal.pcbi.
1013931

**Data availability statement:** The EEG data are not publicly available as they contain information that could compromise the privacy of the research participants. Data can, however, be shared with scientific collaborators after an amendment to the research ethics permit via the hospital's ethics committee and a data transfer agreement. Data can be requested by contacting the ethics committee at HUS at eettinen.toimikunta@hus.fi. The source code used in analysis of this work is freely available and hosted on GitHub (https://github.com/vernaverna/FABEEG).

**Funding:** This research has been funded by Emil Aaltonen Foundation and Finnish Cultural Foundation (to V.H.), the EU Horizon 2020 research and innovation programme #964220 (AI-Mind) (to S.M.), Helsinki–Uusimaa Regional Council, Sigrid Jusélius Foundation (to R.S.), and Research Council of Finland (#355407 to R.S., #321460 and #355409 to H.R., and Flagship of Advanced Mathematics for Sensing Imaging and Modelling grant #359181 to H.R.). The funders had no role in study design, data collection and analysis, decision to publish, or preparation of the manuscript.

**Competing interests:** The authors have declared that no competing interests exist.

sets. Here, we use probabilistic modeling approach on a large pediatric sleep-EEG data set to find low-dimensional individual fingerprints that generalize across sleep stages. We demonstrate that the assumption of non-independent noise is suitable for multi-channel EEG data, providing a tool for fingerprint analysis in challenging clinical contexts.

## Introduction

The human cerebral cortex demonstrates salient, intrinsic oscillatory activity that can be characterized non-invasively with electroencephalography (EEG) and magnetoencephalography (MEG). This spontaneous activity is altered in many neurological diseases and is therefore frequently studied in both research and clinical settings in pediatric and adult populations [1–3]. Despite the high variability of oscillatory activity patterns between individuals [4,5], many typical EEG or MEG features appear to be very consistent within adult subjects over time. Indeed, recent MEG and functional magnetic resonance imaging (fMRI) studies have identified stable neural patterns within individuals' own data samples, commonly referred as brain fingerprints, among functional connectomes and broad-band oscillatory activation patterns [3,6–13], even generalizing across experimental tasks [7,9,12]. Some of the individual variation has been linked to genetic factors [12,14–16], but as the stability of individual features decreases with aging [10] and disease [17,18], individual patterns appear not exclusively determined by genetics. Compared with adults, children and adolescents exhibit higher intra-subject variability and consequently reduced distinctiveness in their connectome profiles [19,20] and oscillatory activity [21], a pattern that may stem from increased neuroplasticity [20]. A delay in reaching adult-like distinctiveness of the connectome can be indicative of poorer mental health [19] or psychiatric disorders, including schizophrenia [22]. As such, stability of brain fingerprints may be a useful metric when studying normal and pathological neurodevelopment [19,20].

Significant changes in oscillatory M/EEG activity appear during brain maturation [23]. Low frequency (<4 Hz) oscillations decrease in power with age, both during sleep [24] and resting wakefulness [25–27], while activity in higher frequency bands increases, as demonstrated by the emergence of theta (4–10 Hz) and sigma oscillations (12–15 Hz) [28] and by changes in the gamma (>30 Hz) range [29]. In addition, the peak frequency in the alpha-range (8–12 Hz) starts to increase after around 16 months of age [25,29]. Sleep-related M/EEG features, such as spindle density, are also associated with age, although the relationship appears non-linear [30]. While maturation-related changes in EEG patterns generally follow a predictable trajectory, their variability is high between individuals [30–32]. Interestingly, the difference between the chronological age of an individual and age predicted by from their structural and functional neuroimaging features, or the so-called functional brain age, has been connected to cognitive functions both in healthy [33] and diseased populations [34]. The prediction error also seems to decrease as a function of age in pediatric populations [35]. Neuroimaging features thus show potential as cognitive, or brain health, biomarkers. The stability of these individual features, or the lack thereof,

could similarly serve as a surrogate biomarker of a risk for atypical development. However, before examining clinically relevant deviations, the individual stability and variation of neuroimaging features over brain maturation in normally developing children need to be addressed. Furthermore, any approach aimed for clinical use should be validated with real clinical data [36], and its flexibility addressed in the varying and often noisy circumstances encountered in clinical practice.

Sleep EEG offers an excellent measure for studying the effects of individual variation and age in large cohorts of children. It is cheap and easy to administer even for infants and small children who otherwise may be uncooperative during wakeful recordings. It is widely administered in clinical practice, e.g., in epilepsy diagnostics in the pediatric and adult populations [1,2], thus offering large real-world data sets. Sleep EEG is typically divided into rapid eye movement (REM) and non-REM (nREM) stages, the latter constituting three sub-stages (N1–N3) according to the depth of sleep, each having different spectral profiles. Akin to task-related and resting-state paradigms, sleep EEG has been shown to be very stable in adults [37], with some trait-like features reported also in adolescents and teens despite the ongoing neurodevelopmental changes [38].

Here we aimed to address whether within-session brain fingerprints, based on power spectral features in EEG recordings, can be extracted and generalized over sleep stages in children during maturation. For this, we applied a machine-learning based approach to analyze power spectra in clinical sleep-EEG recordings from a healthy pediatric population ranging from infants to adolescents (N = 782). Attempts to explain high-dimensional and correlated functional brain responses give rise to complex but sparse data models. Such problems are increasingly tackled with probabilistic learning and latent variable models [39,40]. One variant is latent-noise Bayesian reduced-rank regression (BRRR) [41,42], which has shown promising results in discovering fingerprint-like features in adults' resting-state MEG data [12,13]. The latent-noise formulation encodes the assumption that noise affects both the target and explanatory variables similarly and thus allows to exploit the correlation structures present in the data. Previous approaches to find individually stable features from functional neuroimaging data have generally relied on correlations between different measurements of the same participants. Correlation approaches can reliably identify individuals within specific subject groups, such as patients and controls, and show meaningful relationships with neurological systems affected by the disease [17]. While correlation-based fingerprints are typically derived within one physiological state using all available data, we sought to determine whether reduced-rank regression could improve performance in more challenging fingerprinting scenarios—specifically across sleep stages and in refining individual feature estimation in noisy clinical settings.

We sought to find the latent representation of the EEG data that would maximally differentiate subjects from each other while preserving individual stability over data samples. The formulated model was validated by using it to differentiate individuals within and between different sleep stages, as well as to classify data from unseen participants. We compared the results to those obtained using a correlation-based method and tested the effect of age and sex on the differentiability of the subjects. We hypothesized that sleep patterns would become increasingly stable with age and thus allow for better individual data matching across sleep stages along with maturation; sleep-EEG patterns become increasingly adult-like during late childhood and adolescence [43].

We demonstrate here high individual variation and within-session differentiability of children using their sleep-EEG bandpower features. The BRRR-based low-dimensional representation resulted in individual fingerprints that were more stable than those obtained through a correlation approach and generalized across sleep stages. In addition, the stability of the low-dimensional fingerprint in light sleep increased with age, an effect that was also discernible in correlation-based fingerprinting.

## Results

### Spectral power patterns change non-linearly during childhood

The dataset consisted of 19-channel clinical EEG recordings of non-REM sleep in 782 healthy children aged between 6 weeks and 19 years (mean age ± SD: 4.6 ± 4.3 years). The analyzed part of the recording encompassed 900 seconds of

spontaneous sleep data. The first third (300 s, categorized as N1) consisted of resting EEG before the onset of N2 sleep, followed by 600 s of EEG after the first signs of N2 sleep. From the EEG recordings, we calculated power spectral density estimates (PSDs) for six data segments: two 60s segments that were categorized as N1 sleep, and four 60s segments that were categorized as N2 sleep.

Age-related differences were observed in the PSD patterns of the N1 and N2 sleep segments (Fig 1B). For both sleep stages, the overall spectral power, as indicated by the area under the curve (AUC) of the power spectrum, changed with age non-linearly (N1 sleep: 2nd degree term, $\beta = -9.92$, $SE = 1.03$, 95% CI=[−11.95, −7.89], $p < 0.001$; see S1 Table). During the first years of life, the total power increased and then started to decline with age. The trends were similar in N2 sleep (S2 Table) and might be partly related to scull ossification. Overall the spectral patterns of sleep varied across different age groups (Fig 1A) and participants (Fig 1C). Oscillatory phenomena emerged with maturation: the N1 and N2 segments within individuals differed from each other due to, e.g., emergence of sleep spindles, beta activity, and presence of low-frequency oscillatory activity (Fig 1C). In all age groups, the sleep stages differed from each other at around 12–15 Hz, i.e., the sleep spindle frequency range (cluster permutation test, $p < 0.01$). Despite the prominent age-related differences in the sleep PSDs, these salient global differences were not always present at the individual level (Fig 1C).

We tested the stability of grand average PSD segments from the same sleep stage using cluster-level permutation tests, and examined the results per age group. Within the N1 sleep stage, there were systematic differences between averaged data segments of subjects in 6/10 of the age groups ($p < 0.01$), especially at frequencies above 25 Hz (Fig 2A). Within the N2 sleep stage, differences also appeared in 6/10 age groups ($p < 0.01$), showing trend-like increases or decreases as the recording progressed in time (Fig 2B). Especially the high-frequency bands seem more unstable within N1 sleep stage, whereas oscillatory patterns around 10Hz emerged as N2 sleep progressed. These results illustrate that fingerprinting even within a sleep stage is not a trivial task: correlation-based similarity measures suffer when, e.g., 10 Hz oscillations emerge to PSDs during the sleep stage.

## Cortical fingerprints in children can be identified from EEG spectral power

To identify individual brain fingerprints that best discriminate between subjects, we applied the BRRR algorithm to features obtained from the PSDs. We first calculated the bandpower over 13 logarithmically widening frequency bands between 1 and 43 Hz separately for each channel (see Fig 3A and 3B). The channel-wise relative bandpowers in each frequency band were then used as features in the BRRR algorithm. Differentiation of subjects was carried out both within and between sleep stages. Due to the evident maturation effects, we applied the BRRR model first to all participants (N = 782; Group 'All') and then separately to school-aged children (N = 216; Group '>7-year-olds') and compared the performance and latent space structures of the models. We hypothesized that the model trained using older participants could provide more stable across sleep stage fingerprints compared to the model based on all participants.

In the BRRR model, the regression equation for the measured EEG data (see Fig 3B and Eq 1 in Modeling), encodes the assumption that noise and covariates affect the response via a shared subspace $\Gamma$ and approximates the response using a low-rank decomposition. The model aims at finding the loadings $\Gamma$ that effectively maximize the distance between subjects while minimizing the within-subject distance (Fig 3C and 3D). For feature extraction, a low-dimensional representation of the data is useful. While most of the information from the original data is retained, the low-dimensional representation refines the relevant features for the machine learning objective without sacrificing the model performance. Here, a low-dimensional representation of the data was obtained by enforcing a latent space dimension smaller than both the number of features ($s$) and observations ($p$), i.e., $k = 30 \ll p, s$. This is achieved by imposing shrinkage priors to $\Omega$ and $\Gamma$ (For details, see [41]). The dimension of the latent space was defined using the elbow-method (see Methods and Fig 8A).

To evaluate how well the model differentiated between subjects, we computed the L1 distances between test subjects' EEG bandpower segments which were projected into the low-dimensional latent component space produced

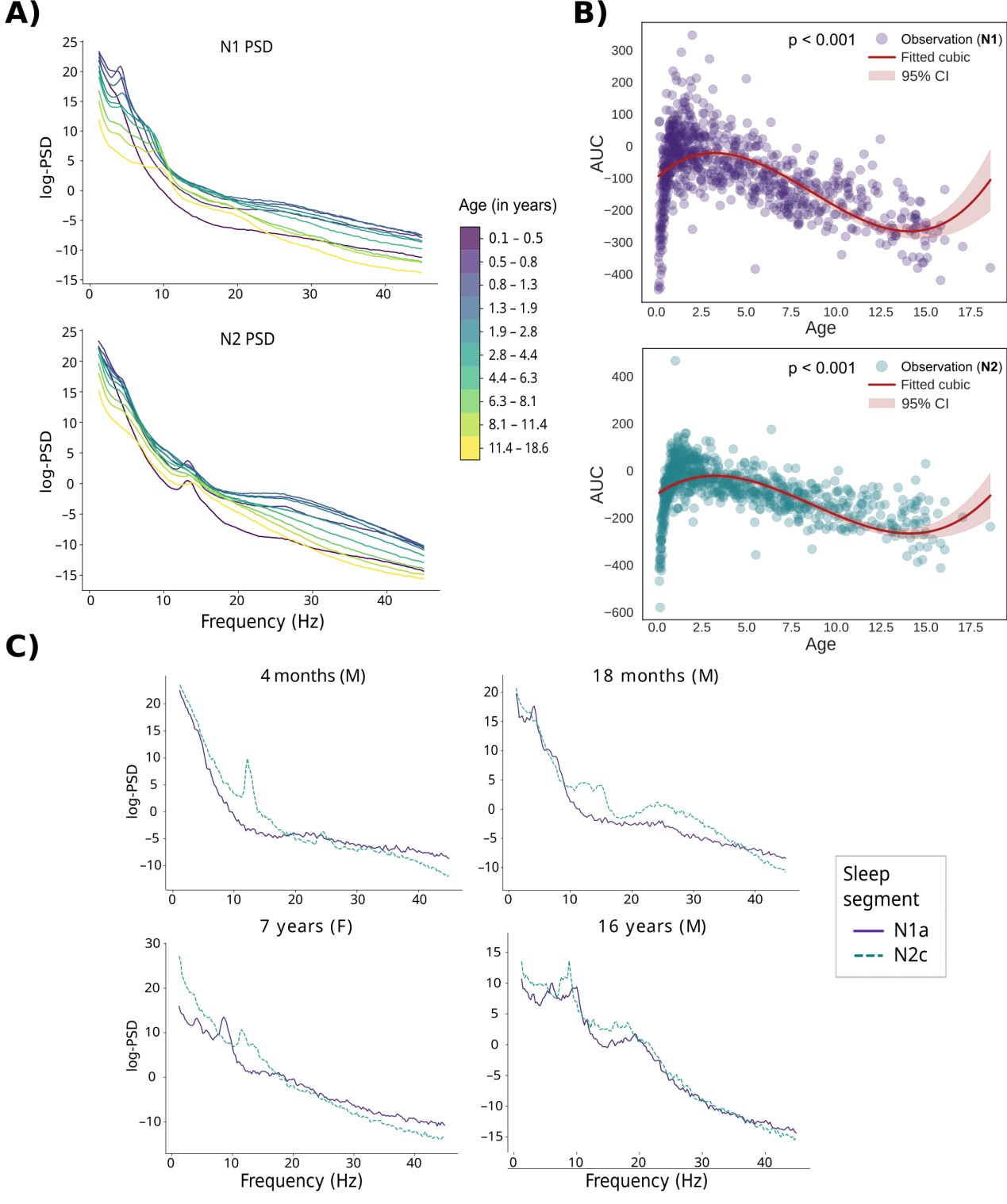

**Fig 1. Age relationships of sleep PSD features. A)** Grand average PSD estimates based on 1-minute segments of N1 and N2 sleep, calculated for ten equal-sized age groups. PSDs are represented on a logarithmic scale. **B)** The N1-PSD area under curve (AUC) demonstrates the increase in the total spectral power during the first years of life and the following decrease in both sleep stages, here illustrated using cubic regression on age. **C)** PSDs averaged over all channels in four example subjects across 1-min sleep state segments (one from N1 sleep stage, one from N2 sleep stage). F= female, M = male. N1a and N2c refer to the first and third segments within the respective sleep stages.

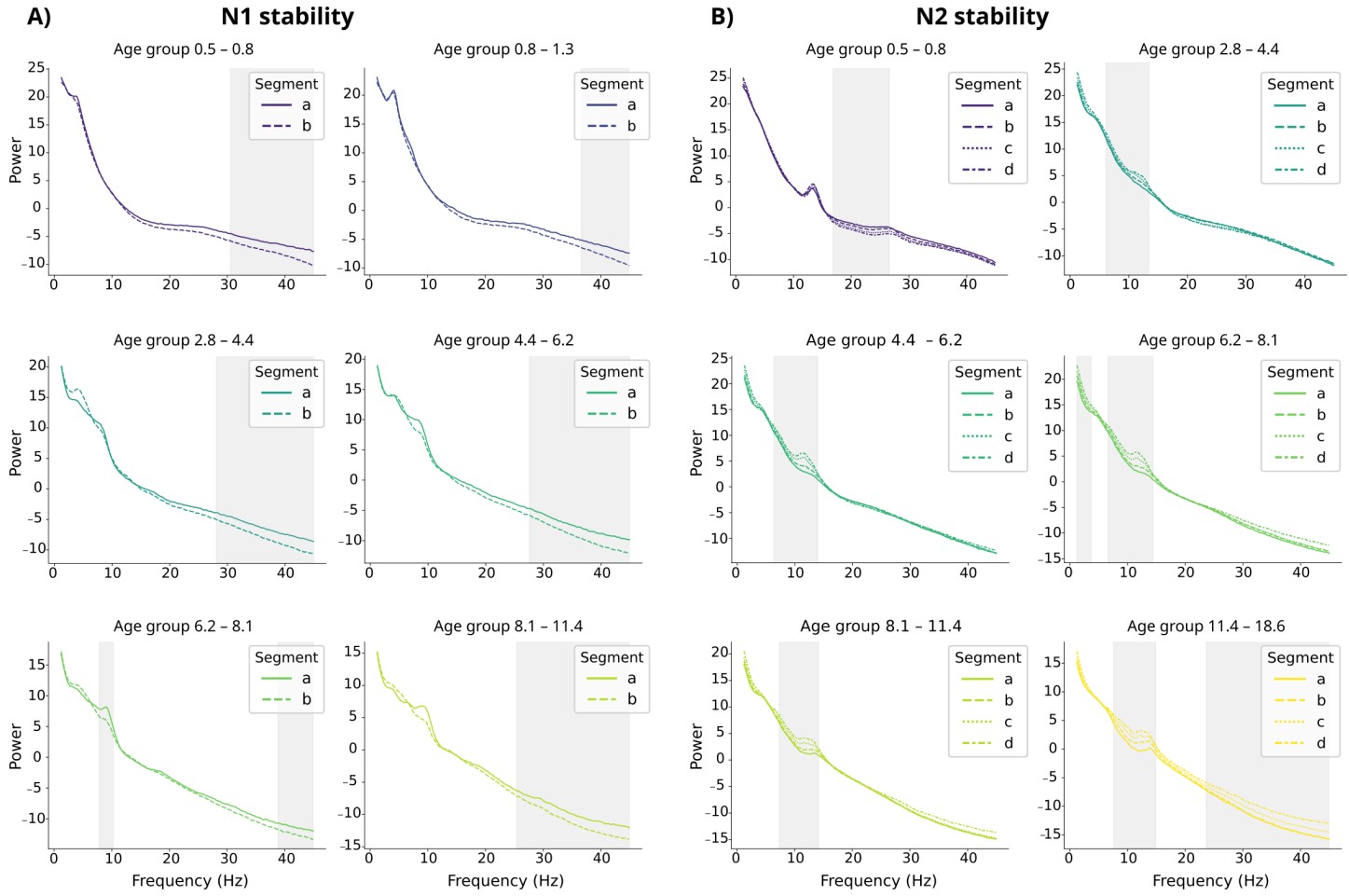

**Fig 2**. **Stability of the average PSD data within sleep stages.** The PSDs were averaged over all channels. Result are visualized for average segments across age groups during N1 (**A**) and N2 sleep (**B**). Note the different age groups in Figs A and B. Clusters indicating significant differences ($p < 0.01$) between segments within sleep stages are highlighted. The segments are marked chronologically (a–b, a–b–c–d).

by the BRRR model. A subject was considered correctly identified if the minimum distance over any set of distances occurred between the individual's own data segments (see Fig 3C). The success rate of the BRRR model was estimated using 10-fold cross validation (CV) as the percentage of correctly differentiated test subjects [7]. We also calculated the success rates by using all the observations in different data segment combinations. To examine the stability of the EEG fingerprint over the course of the sleep recording, we tested how well the latent representation of the BRRR model generalizes to subsequent PSD segments of the same individuals. Ideally, the model should assign each new observation to the correct individual by preserving short within-subject distances in the latent space learned from prior data. For this, we used an additional validation procedure, where the distances between subjects were calculated using additional PSD segments of N1 or N2 sleep, projected into the latent space. The BRRR fingerprinting model was bench-marked against a correlation-based similarity measure, which is commonly used in functional brain-fingerprinting [8,11,44]. Unlike in BRRR, the correlations between observations were computed on the full data matrix **Y** (Fig 3D) without any dimensionality reduction. For both BRRR and correlation methods, we defined the differentiability as a Z-scored self-distance (or correlation, when using correlation method) against the distances (correlations) to others. The Z-scoring approach is described in the Methods section (Eq 3, Statistical testing).

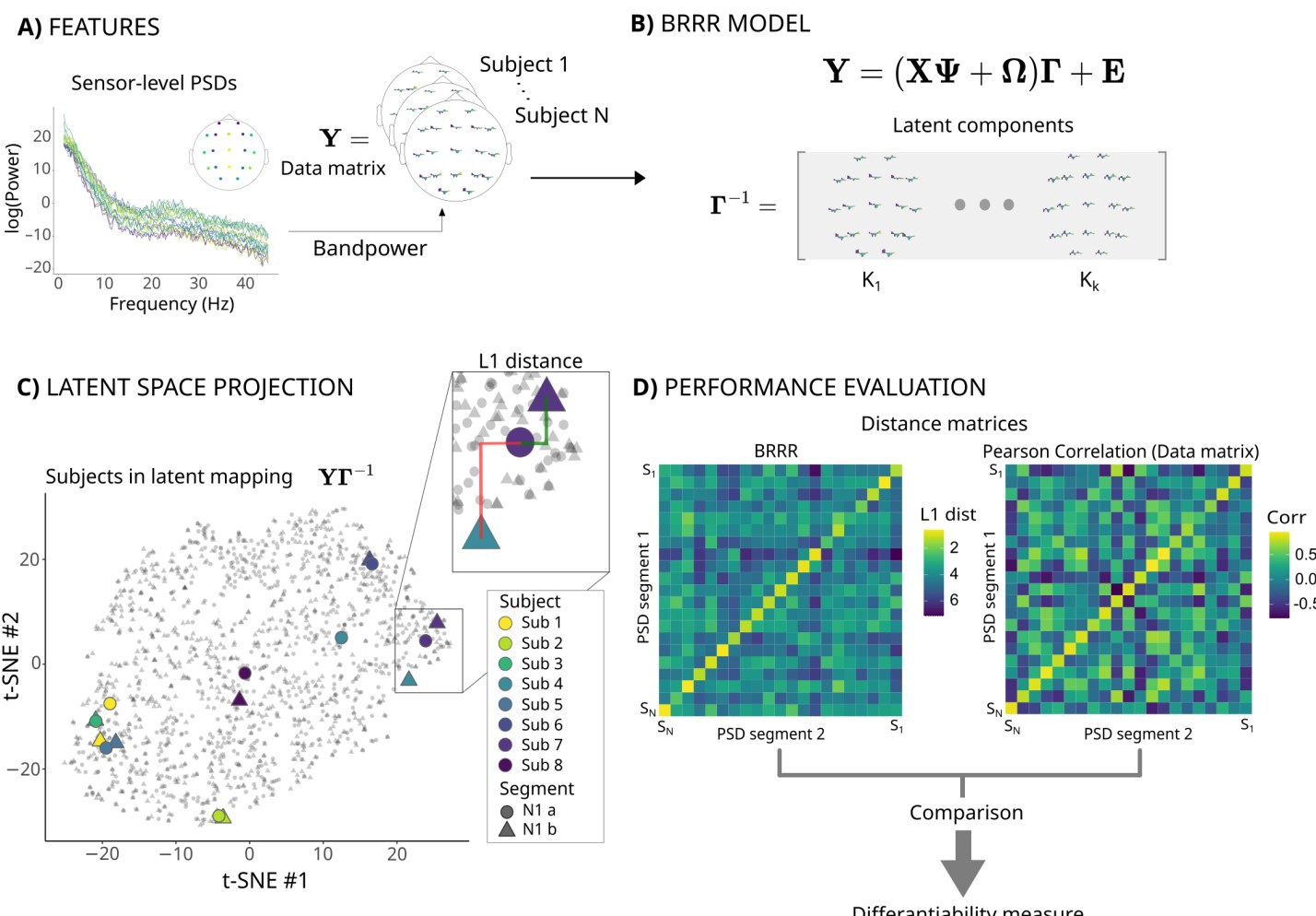

**Fig 3. Overview of the analysis pipeline.** We calculated, for each subject, the sensor-level PSD estimates (**A**) which were then vectorized and concatenated into a data matrix. The individual bandpowers in different sleep stages were used as observations in the BRRR model (**B**) which produces a low-dimensional representation that best differentiates between the subjects (here the loading matrix $\Gamma$). Each subject was then projected onto the low-dimensional space produced by BRRR (**C**), here visualized, for simplicity, in two dimensions using stochastic neighborhood embedding (tSNE). The distances between subjects' own data projections (within-subject distance) were compared to the distances to other subjects (between-subject distances). These distances were collected into similarity matrices (**D**) from which the overall fingerprinting success rates and differentiability scores (Z-scored self-distances) were calculated. These metrics resulting from application of the BRRR model were further compared to fingerprinting results based on Pearson correlation of the full-dimensional data matrix **Y**.

**Within sleep stage models result in good fingerprinting accuracy.** We first conducted the fingerprinting analysis with BRRR using 10-fold cross-validation. For both subject groups ('All' and '>7-year-olds'), test subjects could be differentiated with over 87% success rate (88% for All, 87% for > 7-year-olds) when spectra from two N2 segments were used as observations in the training, and with over 73% success rate when two N1 data segments were used (see Fig 4A; For more information, see S3 Table). The N2 model explained up to 85% of the variation in the response (proportion of total variance explained, PTVE), indicating that most of the variation in the EEG bandpower within sleep stage can be attributed to the individuals. Adding a third N2 segment to the training data slightly increased the success rate for group 'All' (90%), while remaining the same for '>7-year-olds', again demonstrating good stability of the low-dimensional fingerprints. The model fit remained high at 80%.

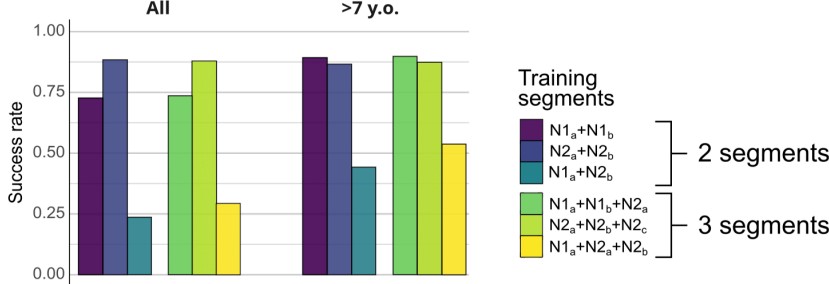

**A)** 10-fold CV results from BRRR

**B)** Fingerprinting success rates accross age groups and sleep stages

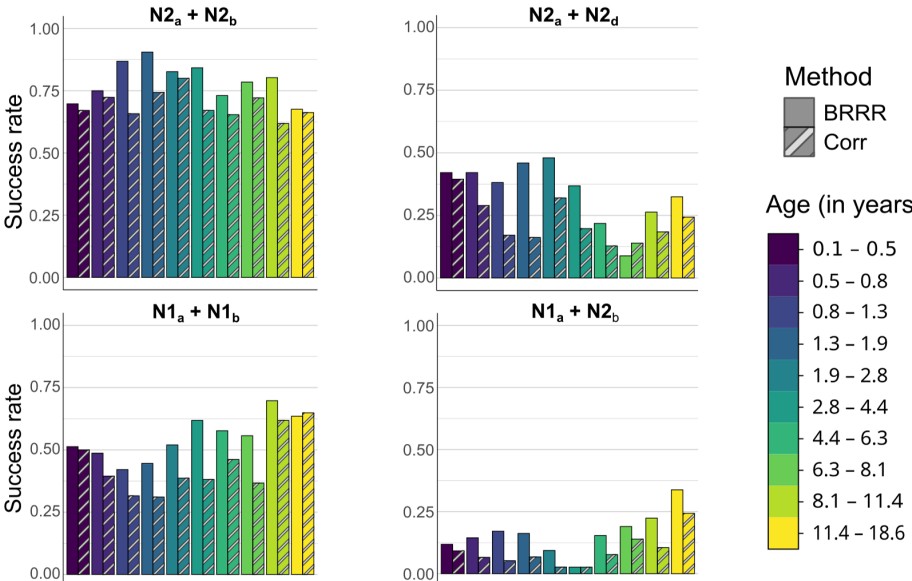

**C)** Differentiability score distributions in different fingerprinting tasks

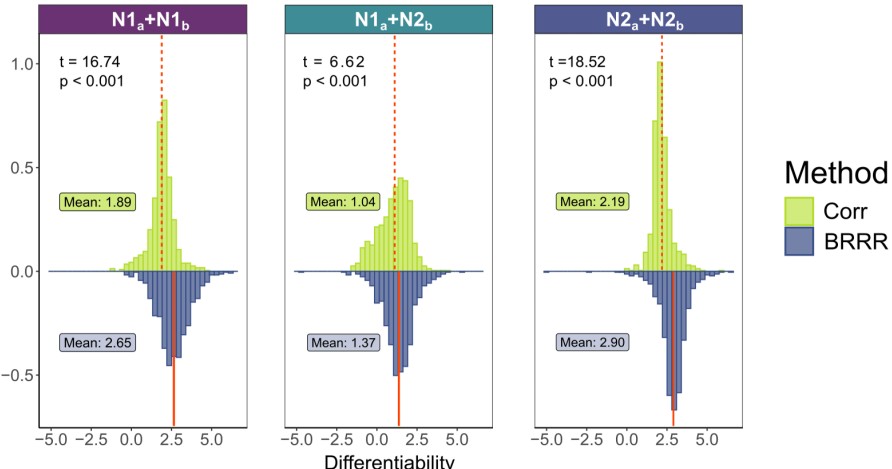

**Fig 4. Success rates and differentiability scores across different models and subject groups. A**: Averaged 10-fold cross-validation success rates for BRRR models using 2 or 3 training data segments. Success rates are shown separately for differentiating subjects in groups 'All' and '>7-year-olds'. **B**: Fingerprinting success rate on all data per age group, for both BRRR and correlation. Upper row: fingerprinting within N2 sleep, either using subsequent segments (left) or segments further apart in time (right). Bottom row: fingerprinting results within N1 sleep (left) and between sleep stages (right) **C**: differentiability score distributions in three fingerprinting tasks for BRRR and correlation -based fingerprinting. The means of the differentiability score distributions are indicated by vertical lines and compared using t-test.

For the N1 sleep, the models trained with data from over 7-year-olds performed better than the models trained with all data: using two N1 data segments as training data yielded differentiability scores 89% and 73%, respectively (Fig 4A and S3 Table). Generally, models containing observations from both sleep stages did not perform as well as models trained within sleep stage. When the training data contained two N1 data segments and one N2 segment, nearly 90% success rate was reached for '>7-year-olds' and 72% success rate for the full dataset. However, the success rate for subject differentiation dropped below 30% for 'All' when the training data included only one N1 data segment combined with one or two N2 data segments, probably related to more heterogeneous input data. For '>7-year-olds', the success rates were higher (44–54%) than for models trained on the full dataset, although clearly lower than within-sleep stage success rates. Although the additional variance associated with sleep stage properties could no longer be predominantly attributed to individual differences (as indicated by the decrease in the model fit), the BRRR model still accounted for most of the variation in the response (PTVE >56%).

We then computed the fingerprinting results on whole data (without cross-validation and using all subjects) with both BRRR and correlation-based fingerprinting on two data segments per individual. The results from both methods were in agreement: best success rates were obtained within N2 sleep stage and worst when using mixed sleep stages (S4 Table). Notably, BRRR had higher success rate (approx. 10%) than correlation over the various fingerprinting tasks. Fingerprinting was more successful when the data segments were closer to each other in time (fingerprinting on $N2_a + N2_b$: 80% vs. on $N2_a + N2_d$: 36%, S4 Table). When examining the success rates per age group, we found different patterns depending on the fingerprinting task (Fig 4B). Within subsequent segments of N2 sleep (upper left), the success rates were high in all age groups, but seemed to increase by age in N1 sleep and mixed sleep stage ($N1_a + N2_b$) fingerprinting (bottom row), corresponding to results for over 7-year-olds in Fig 4A. When fingerprinting on far-apart N2 segments, the older age groups did not appear to be better differentiated from the rest. Overall, BRRR and correlation show similar trends for the fingerprinting success rate across age groups, with some exceptions for the $N2_a + N2_d$ fingerprinting task, where BRRR achieved highest success rates for children aged $1.3 - 2.8$ years.

The differentiability score distributions reflect the success rate trends for different fingerprinting tasks (Fig 4C). The average differentiability scores were highest for N2 fingerprinting for both BRRR and correlation and lowest for the mixed fingerprinting task using data from both sleep stages ($N1_a + N2_b$). BRRR performed better (higher differentiability scores) than correlation in all depicted fingerprinting tasks, although the average differentiability scores in the mixed sleep stage fingerprinting task between correlation and BRRR were fairly similar (mean Diff. for BRRR=1.37, mean Diff. for correlation=1.04).

**BRRR can produce generalizable fingerprints.** We also tested how well the latent representation learned by BRRR using two data segments could generalize to other data segments. Here again we examined separately models trained with groups 'All' and '>7-year-olds'. The results are summarized in S5 Table. Within N2 sleep, the learned latent representation generalized well to other N2 segments and the results were comparable for both subject groups. The latent space learned from $N2_a + N2_d$ segments managed to differentiate subjects based on their $N2_b + N2_c$ segments with success rates 0.8 ('All') and 0.76 ('>7 y.o.'). Notably, the model trained with far apart segments had better generalization results than when trying to fingerprint based on those segments alone (Fig 4B, top right). For over seven year olds, the N2 latent representations were successful when fingerprinting in N1 sleep (success rates $0.72 - 0.76$), whereas for all subjects, the success rates were lower ($0.48 - 0.54$). The mixed sleep latent space generalized well to N2 segments (SR=0.74, 'All') but the reverse was not true (SR=0.11, 'All'). We also examined the success rates separately for over seven year olds in the models trained using all data (S5 Table). The success rates were always higher than in the whole population, but when trying to generalize from the N2 to N1 sleep stage, the success rates were lower ($0.58 - 0.63$) compared to models trained exclusively with over 7-year olds. Here, restricting the analysis to older participants may allow the model to find more generalizable features.

The success rates based on fingerprinting with new segments were in most cases higher than the correlation-based fingerprinting on the same segments (S4 and S5 Tables). This indicates that BRRR could find individual features that

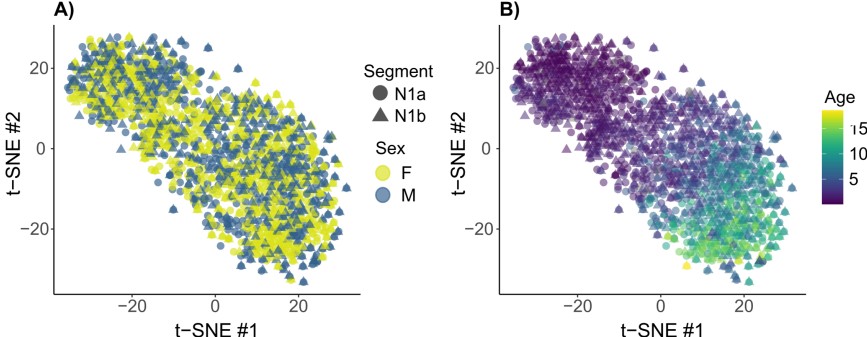

were more general than those solely related to training data. Overall, the models trained with more heterogeneous input data generalized to segments that are temporally close together or more similar (e.g., $N1_a + N2_d$ latent space projection manages to fingerprint on $N2_a + N2_b$ segments with 74% success rate (S5 Table).

The correlation-based (dis)similarities between subjects when fingerprinting on $N2_a + N2b$ were very alike to between-subject dissimilarities when fingerprinting on $N2_c + N2_d$ -segments (Mantel's test on dissimilarity matrices, $r = 0.69, p < 0.001$). For BRRR, the association between similarity matrices was weaker across N2 sleep segments (Mantel's $r = 0.56, p < 0.001$). Between N1 ($N1_a + N1_b$) and N2 ($N2_a + N2_b$) fingerprinting, the dissimilarity structures between subjects were less alike for both fingerprinting approaches (Correlation: $r = 0.48, p < 0.001$; BRRR: $r = 0.44, p < 0.001$). Lower values indicate that the pattern of between-subject dissimilarities changes more when fingerprinting using N1 sleep compared to N2 sleep. Despite this, comparing the correlation and BRRR dissimilarity matrices showed similar levels of association in both N1 fingerprinting ($r = 0.54, p < 0.001$) and N2 fingerprinting ($r = 0.51, p < 0.001$). Interestingly, in $N2_a + N2_d$ fingerprinting, the association of BRRR and correlation-based dissimilarity matrices was weaker ($r = 0.32, p < 0.001$), despite both methods showing very stable dissimilarity structures when comparing $N2_a + N2_b$ to far-apart $N2_a + N2_d$ -fingerprinting dissimilarities (BRRR: $r = 0.73$, Corr: $r = 0.77, p < 0.001$). It therefore appears that BRRR and correlation-based fingerprinting exploit slightly different features in the data when the sleep segments are further apart, resulting in different between-subject dissimilarity maps.

## Differentiability is affected by age

As suggested by the within N1 sleep and mixed sleep stage results above, the stability of individual features across sleep stages appeared somewhat age dependent. Indeed, age emerged as an organizing factor in the latent space learned from N1 data segments, as illustrated in the two-dimensional t-distributed stochastic neighborhood embedding (t-SNE; [45]) map in Fig 5B. The sex of the participants, however, did not similarly separate the subjects in the latent space (Fig 5A). Similar projections for N2 sleep model can be viewed in S1 Fig; here, too, we observed organization according to age but not sex.

We examined how individual differentiability in population is affected by variables such as age, sex and possible confounders in the data. To this end, we fit a linear model to test the effect of age, sex, ratio of muscle artifact and cap size (factor variable) to the differentiability scores obtained from BRRR method and correlational fingerprinting. We also tested possible interactions with muscle artifact-levels and sex.

**Fig 5. Subjects in N1 latent space projection.** Two-dimensional projection of the subjects' latent spaces (2N1 model trained with one N1 data segment and tested on another N1 data segment), coloured by sex (**A**) and age (**B**) and visualized with t-distributed stochastic neighbor embedding. The first N1 segment is marked with a circle, the second with a triangle. The age of the participants emerges as an organizing structure in the latent space while the sex does not.

PLOS Computational Biology

For N2 differentiability, we found no significant relationships for BRRR (Fig 6A), but a slight negative effect of larger cap-size was observed for correlation (Cap[FT]: $\beta = -0.27$, $CI_{95\%}$=[$-0.44$, $-0.09$], $p = 0.0034$). The trends of the differentiability scores for girls and boys seemed to be similar, given that sex or muscle artifact-sex interaction terms were not significant (Fig 6A), but the result might attributable to outliers given the high uncertainty in the estimates. The full model details are given in S6 Table. Despite the age gradient in the latent space (S1 Fig), differentiability scores had no age relationship.

In contrast, the N1 differentiability showed a significant positive age effect for BRRR ($\beta_{\text{Female}} = 0.11$, $CI_{95\%} = [0.07, 0.15]$, $p < 0.001$) and correlation ($\beta_{\text{Female}} = 0.05$, $CI_{95\%} = [0.02, 0.08]$, $p < 0.001$), as detailed in Fig 6B and Table 1. The difference between muscle artifact levels of N1 fingerprinting segments were not associated with differentiability for either

## A) Multiple regression on N2$_a$+N2$_b$ differentiability

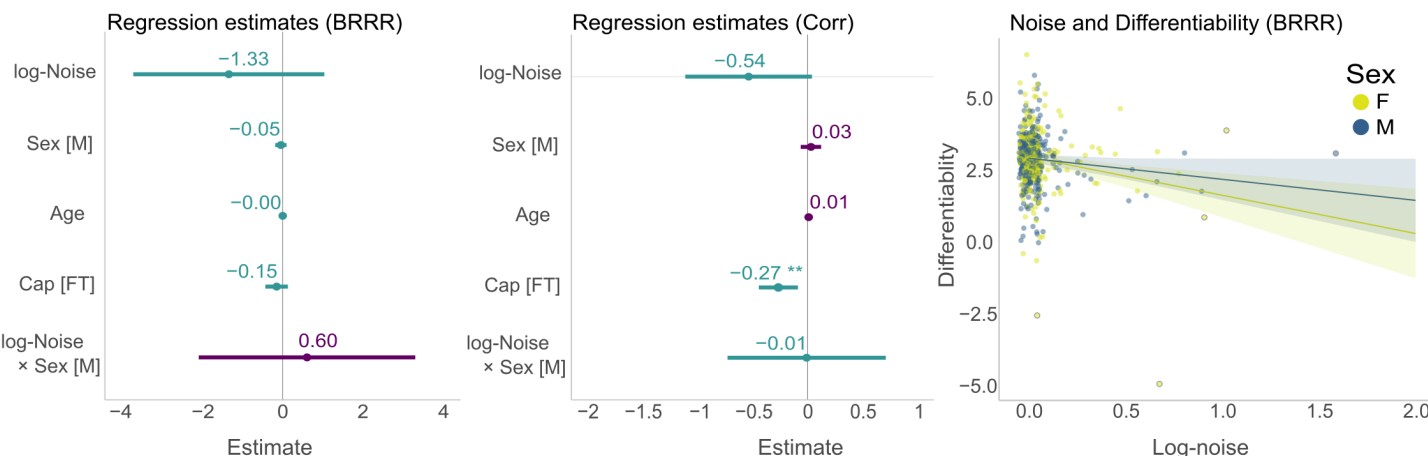

## B) Multiple regression on N1$_a$+N1$_b$ differentiability

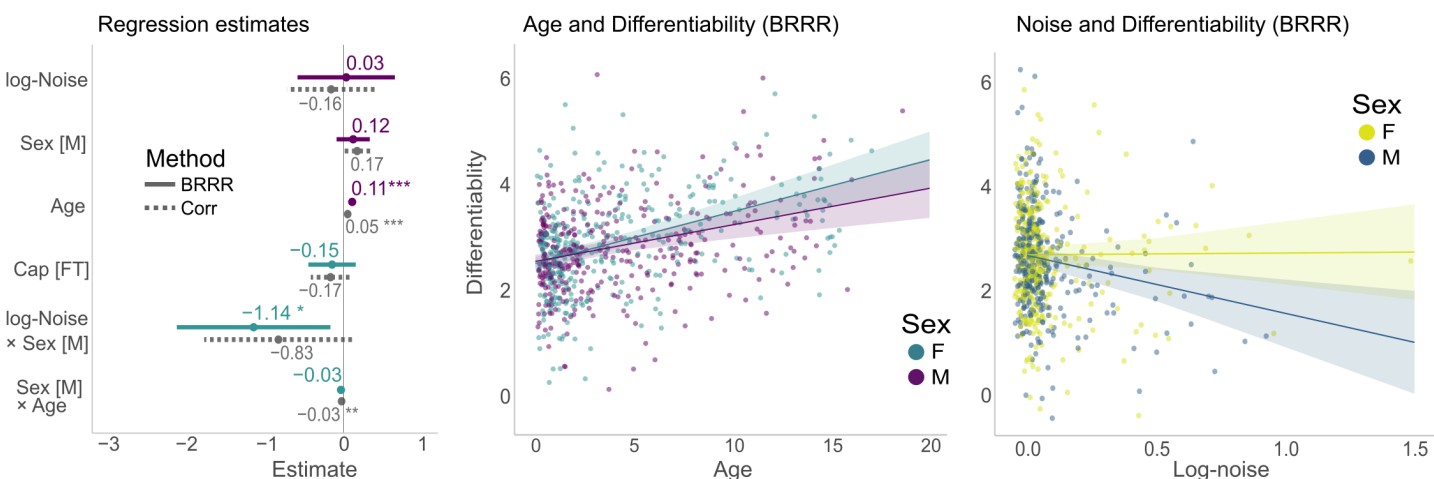

**Fig 6. Multiple regression on differentiability scores. A:** Regression estimates for variables using N2 differentiability scores obtained from BRRR (left) and correlation (middle). The interaction effect of age and sex is plotted for BRRR (right). **B:** Regression estimates for N1 differentiability scores for both fingerprinting methods (left). The marginal age effect (middle) and noise (muscle artifact)-sex interaction effect (right) with 95% confidence intervals are visualized for BRRR only. Statistically significant effects are marked with asterisks: (***)$p < 0.001$, (**)$p < 0.01$, (*)$p < 0.05$.

**Table 1**. Regression results for BRRR and Correlational differentiability scores from N1 sleep.

| | BRRR | | | | | Correlation | | | | |
|---|---|---|---|---|---|---|---|---|---|---|
| | Coef. | SE | $t$ | CI [0.025 | 0.975] | Coef. | SE | $t$ | CI [0.025 | 0.975] |
| Intercept | 2.27 | 0.07 | 30.73*** | 2.12 | 2.41 | 1.75 | 0.06 | 27.26*** | 1.63 | 1.87 |
| noise | 0.03 | 0.32 | 0.11 | −0.58 | 0.65 | −0.16 | 0.28 | −0.57 | −0.71 | 0.39 |
| sex[M] | 0.12 | 0.11 | 1.14 | −0.09 | 0.33 | 0.17 | 0.09 | 1.96 | 0.00 | 0.34 |
| age | 0.11 | 0.02 | 5.89*** | 0.07 | 0.15 | 0.05 | 0.01 | 3.76*** | 0.02 | 0.08 |
| cap[FT] | −0.15 | 0.15 | −0.95 | −0.45 | 0.16 | −0.17 | 0.13 | −1.34 | −0.42 | 0.08 |
| noise:sex[M] | −1.14 | 0.50 | −2.30* | −2.12 | −0.17 | −0.83 | 0.48 | −1.74 | −1.77 | 0.11 |
| sex[M]:age | −0.03 | 0.02 | −1.73 | −0.07 | 0.00 | −0.03 | 0.01 | −2.58** | −0.05 | −0.01 |
| **Model Statistics** | | | | | | | | | | |
| Residual Std. Error | 0.919 | | | | | 0.742 | | | | |
| Multiple $R^2$ | 0.143 | | | | | 0.042 | | | | |
| Adj. $R^2$ | 0.136 | | | | | 0.034 | | | | |
| $F$-statistic (6,705) | 19.68, $p = 2.6 \times 10^{-21}$ | | | | | 5.19, $p = 3.1 \times 10^{-5}$ | | | | |

Here N1 sleep contains segments $N1_a + N1_b$. Significant effects are marked with an asterisk: ***)$p < 0.001$, **)$p < 0.01$, *)$p < 0.05$.

of the approaches. For BRRR, the boys' differentiability scores decreased more when muscle activity levels increased ($\beta_{\text{log-Noise:Male}} = -1.14$, $CI_{95\%} = [-2.12, -0.17]$, $p = 0.022$; Fig 6B and Table 1). For correlation based differentiability scores, the girls' differentiability improved more steeply as a function of age ($\beta_{\text{Male:Age}} = -0.03$, $CI_{95\%} = [-0.05, -0.01]$, $p = 0.01$; See also S2 Fig). When differentiability scores were determined from the $N1_a + N2_b$ fingerprinting tasks, we did not observe any significant age, sex or muscle artifact effects for either of the fingerprinting approaches. The detailed results of that regression analysis are given in S7 Table.

We also tested separately if artifact load changed with age using linear regression. There the ratio of muscle artifacts seemed to slightly decrease with age in all of the combinations above: in N2 sleep, the effect was smallest ($\beta = -0.0048$, $CI_{95\%} = [-0.0068, -0.0028]$, $p < 0.001$, $r^2 = 0.026$), but the effect of muscle activity was minimal also in N1 sleep ($\beta = -0.0076$, $CI_{95\%} = [-0.0099, -0.0054]$, $p < 0.001$, $r^2 = 0.045$) or in a mixture of N1 and N2 sleep ($\beta = -0.0098$, $CI_{95\%} = [-0.013, -0.0070]$, $p < 0.001$, $r^2 = 0.054$). The relationships are visualized in S3 Fig. As the effect sizes were very small, we did not believe that the age effect on differentiability was due to decrease in muscle artifact changes in N1 sleep. However, the variation in muscle artifact ratio seems to decrease as age increases.

BRRR and correlation-based fingerprinting demonstrated high agreement on differentiability. Neither method wasaffected by muscle activity when fingerprinting within N2 sleep or mixed sleep, but muscle artifact levels in N1 sleep had negative effects on BRRR differentiability for boys. Correlational differentiability increased more rapidly for girls as a function of age. Sex or cap size (except for correlation in N2 fingerprinting) were not related to individual differentiability scores in other sleep stages.

## Discussion

Our individually unique brain structures allow for identification based on anatomical brain features [46], much like the idiosyncratic patterns in fingertips. In recent years, also functional fingerprinting based on individual variation, applied both in M/EEG and fMRI, has gained momentum: especially in adult populations, features like functional narrow-band and broad-band connectome and power spectra have demonstrated fingerprint-like qualities when comparing to reference data from the same individual [11–13,47]. In this study, we examined the possibility of EEG-based fingerprinting using two different approaches in a large pediatric population (N=782) and assessed the stability of broad-band power features within and across N1 and N2 non-REM sleep stages.

## Individual fingerprints generalize across sleep stages

Exploiting the correlation structures in the EEG broad-band power using BRRR yielded a latent-variable representation of the data that could differentiate the individuals from each other and predict the identity of a test subject with up to 90% average success rate. Furthermore, individuals explained most of the variance in sleep EEG bandpower in the different modeling settings. The BRRR models trained within one sleep stage had the highest success rates, which was also true for correlation-based fingerprinting. Unsurprisingly, the explanatory power of the BRRR approach, measured by PTVE values, was highest for these models where non-individual related variance was not increased by adding data segments from different physiological states. Still, the model could fit into individual data very well across sleep stages, demonstrating the robustness of BRRR in conditions with different noise profiles.

Despite the different spectral properties of N1 and N2 sleep stages, the BRRR algorithm could find a latent representation from their combination that allowed for excellent identity prediction up to 90% from test subjects. The mixed sleep-stage models showed lower success rates when fingerprinting included all subjects, compared with analyses limited to participants older than seven years old. The differentiability scores in the latent space mirrored the model success rates, being highest for models trained within N2 sleep and lowest for mixed sleep stage models. When segments used in fingerprinting were far apart in the recording, the success rates were lower, even if they were from the same sleep stage. On the other hand, latent representations learned from these segments generalized well when fingerprinting other data segments that were closer together. We did not achieve as high fingerprinting accuracy scores as reported in some studies using PSD features extracted from MEG data [11–13]. Part of this result might be attributable to low-density EEG used here with relatively granular bandpower features, which may fail to capture all dynamic fluctuations in the brain oscillations. Also in N1 fingerprinting, the amount of muscle artifacts affected the differentiability scores negatively, which likely decreases success rates. However, it has been reported that connectome-based [19,20] and power-spectral based [21] differentiability improves with age as children mature. Thus, the infant-skewed age distribution might result in lower fingerprinting accuracies due to physical and functional maturation effects discussed in more detail below.

We observed systematic differences in the power spectral densities both between and within the sleep stages: for example, the power in the sleep spindle range (12–15 Hz) increased within N2 sleep across subsequent segments. The differences within sleep stage highlight that in longer sleep recordings, fingerprinting approaches within one sleep stage does not guarantee high accuracy. Given the non-linear nature of change in total power levels and considerable individual and sensor dependent variation in the oscillatory power, related e.g., to cap placement, skull thickness and cranial ossification, we opted to use relative bandpower measures over absolute bandpower. Furthermore, skull ossification appears to have only moderate association with certain frequency bands, such as EEG alpha power [48]. Using relative bandpower does not, however, erase all the systemic differences in oscillatory power that were observed within and between the segments, such as power variations at the typical sleep spindle frequency range.

Fingerprints within one physiological state and fingerprints across states have slightly different interpretations. Within-state fingerprints describe the stability of individual features in one condition, and they have been shown to diminish in e.g., Parkinson's disease [17,49]. Across-task fingerprints can be confounded by non-electrophysiological factors that are not meaningfully related to brain function. In our study, across sleep-stage BRRR-fingerprints were robust against muscle artifacts and cap size. Especially within recording session, across-state fingerprints might be more robust against certain types of noise that could spuriously improve differentiation (such as heart beat, muscle or ocular artifacts). Previous research using the BRRR method has revealed that the PSD components that differentiate between sibling pairs also differentiate between individuals, even across measurement tasks, and that some of these components are associated to SNPs [12]. Generalizable fingerprints can reveal trait-like features in brain function which can be correlated to cognitive or clinical measures. Both trait- and state effects have been shown to be important in individual connectivity [50]. A recent MEG study on sibling pairs showed that components of oscillatory somatomotor activity have variable heritability and thus likely reflect different underlying generation mechanisms, some depending on more fixed anatomical parameters and

others reflecting dynamic functional characteristics [51]. In summary, we believe that the different electrophysiological features that drive the differentiation between individuals are likely to reflect both anatomical and functional factors that are not easy to separate from each other.

### Age affects the differentiability of subjects

We had hypothesized that the stability of the fingerprint-like latent variable representations would increase as children mature, as shown by previous literature in connectome-based fingerprinting [20]. In terms of success rates, when predicting the identity of a segment pair using a representation learned from another sleep stage, concentrating the analysis to school-aged participants sometimes improved fingerprinting results, both for BRRR and the correlation-based model. Furthermore, we observed age-related linear trends in the self-differentiability scores. In our analysis, differentiability scores determined from N1 sleep and the mixture of N1 and N2 sleep representations showed the most prominent age effects. The relative bandpower estimates used here reflect in part also aperiodic signals [32,52], which have been demonstrated to be both individual [51,53] and subject to changes during maturation [32]. It is therefore possible that the age-effects observed in the low-dimensional space reflect changes in aperiodic power, a result that has been observed in fingerprinting studies before [21]. Such changes may include changes related to skull-thickening and ossification during the first year of life - aperiodic power levels have been shown to increase most during the first year of life [54]. Furthermore, flattening of the aperiodic activity and slowing of alpha oscillations continue across the human lifespan [55]. We observed subtle age-related differences in sleep behavior, as the differences in muscle activity patterns slightly decreased with age. This did not appear to be reflected in differentiability scores, however. Previous studies have demonstrated that fingerprinting based on oscillatory activity seems to be robust against many non-brain artifacts [17,56].

We observed significant sex effects on differentiability only within N1 sleep. Sex mediated the differentiability scores slightly differently based on the used approach: for BRRR, boys' fingerprint stability was more affected by muscle artifacts than girls', whereas for correlation, the fingerprint stability for girls increased more rapidly than for boys as a function of age. In all sleep stages, large muscle artifacts were relatively rare across all subjects, thus making the regression fits more sensitive to outliers. In a longitudinal adolescent population study, Candaleria-Cook et al. reported significant age-related changes in relative EEG power specifically in males [57]. Connectome-based fingerprints have reported to show sexual dimorphism in adolescence, where distinctiveness of connectome stabilizes earlier in females [20]. Our results seem to align with this result for N1 sleep. It is important to note that the skewed age distribution in our sample increases the uncertainty of the regression fits in the older age brackets and might therefore obscure sex-specific maturation effects in adolescence and in other sleep stages.

### Correlation and latent noise approaches in modeling individual differences

Earlier results on the usefulness of data-reduction techniques for fingerprinting have been mixed, some reporting no improvement in fingerprinting success rates [11] while others claim considerable benefit [8,58]. As opposed to performing subject differentiation based on full feature matrices, we opted to use a latent variable approach, leaning into the intuition that not all features are informative nor independent due to spatial correlations in the fingerprinting task [12]. Furthermore, using the individual identifiers as variables helps to ensure that the low-dimensional space maintains features informative of individuality, as opposed to unsupervised dimensionality reduction methods. To ensure the soundness of the method in this cohort, we compared our results to a correlation-based fingerprinting approach.

Fingerprinting success rates benefited from informed regularization: the BRRR approach did not suffer as much from increasing the number of subjects as the correlation-based method did. However, the BRRR model was sensitive to overfitting: including too many components reduced the model's fingerprinting performance drastically, while the amount of explained variance by the model remained stable. Latent space dimension, which is a hyperparameter, needs to be set separately for different applications. In contrast, correlational fingerprinting is computationally simpler and does not require

hyperparameter optimization, and the method has given excellent results in the context of individual connectomes [7,8] and PSDs [11,13]. As a model, the BRRR approach can be trained on a set of subjects and applied to another dataset, which can help to assess the consistency of neurofunctional interindividual differences across data sets: this has been demonstrated in the context of resting-state connectomes [13].

Generally, subject differentiation in 30-dimensional latent space provided by BRRR outperformed the correlation-based differentiation on the full 247-dimensional feature matrix, where success rates were around 0.1 units lower. On average, the differentiability scores obtained with the BRRR method were also higher. When examining the success rate distributions in terms of age groups, we noticed that the correlation-based and BRRR success rates were closer to each other in <6 months olds and in over 11 year old subjects. The below 6 months old age group had very stable PSD estimates within sleep stages, which might explain the higher results in correlation-based fingerprinting. Older subjects, on the other hand, are more discernible. While BRRR yields better results in terms of differentiation, the structure of the latent space is stochastic, which complicates the physiological interpretation of the components. When the number of required latent components to fit the data is small, together with genetic covariates they may have meaningful physiological interpretation [12] that uncover stable individual features across physiological states. Dimensionality reduction methods in fingerprinting might introduce a trade-off between simplicity and performance. With correlation-based fingerprinting using all data features, it is more straightforward to determine which brain areas or networks and frequency components contribute to individual differentiability [7,49].

The trends in success rates and differentiability scores were similar for both fingerprinting approaches. The strength of the BRRR approach is highlighted when using different sleep segment to train and test the model: latent representations learned from mixed sleep stages generalized to other sleep segments, and had slightly higher success rates than the correlational approach. This is likely due to the low-dimensional representation of the data which reduces noise (including measurement noise and importantly, non-modeled effects) while refining the individual features. Thus, latent mapping can maintain low within-subject distances even when presented with new data. A previous study using BRRR to source-localized MEG found slightly better results with correlation-based fingerprinting than reduced-rank regression [13]: we suggest that in the case of large sample-size sensor-level clinical EEG, the latent noise model is especially beneficial. Furthermore, the BRRR approach allows for using more than two data segments per subject when learning the latent representation, which allows for modeling salient individual features over different physiological states.

When a relationship between age and differentiability was present, both approaches showed significant effect between differentiability and age. In BRRR, the effect size was twice as high. In N2 sleep fingerprinting, larger cap size had significant negative effect on correlation-based differentiability scores. The PSD estimates from this sleep stage were unstable at alpha- or sleep spindle range oscillations in older age groups, which were measured with the larger cap size. Both method seemed to be relatively robust against muscle artifacts in N2 sleep and mixed sleep stages. However, the estimated muscle artifact effects in BRRR differentiability had more uncertainty. This is likely related to the L1-distance measure used in building the dissimilarity matrix: the distances between subjects have a wider range than correlations, which leads to larger variance and/or outliers and thus more variable differentiability scores. As a similarity metric, correlation has a bounded target range. On the other hand, using linear relationship to measure similarity might ignore possible nonlinear associations. Depending on the features used for fingerprinting, utilizing similarity measures based on, e.g., mutual information, could reveal new relationships.

## Brain fingerprints over developmental periods

To establish a long-term EEG fingerprint, the BRRR model should generalize to new observations from the same individuals, preferably over separate measurement sessions. Our results point to stable functional fingerprints in developing children across sleep stages measured within one session, but long-term fingerprints over separate sessions measured far apart have not yet been established.

The short-term test-retest reliability (>1 week) in M/EEG oscillatory activity seems to be good in children [59], but maturation changes the oscillatory activity patterns significantly in the long term in school-aged children [57]. As infants undergo even greater development during their first year of life, including the emergence of rhythmic oscillatory activity [28,60], the functional dynamics in this stage are less likely to be preserved into adulthood. Interestingly, the maturation-related changes in the power spectra seem to be very consistent across individuals during the first two years of life, after which individual differences start to emerge [28]. While the functional fingerprints may not generalize over the developmental period within individuals, it might be a characteristic feature indicating typical development: if maturation occurs normally, the power-spectral fingerprints identified in infancy should fade after certain (yet to be specified) time. Indeed, the short-term reliability in EEG-PSD estimates tends to be higher in typically developing children compared to children with autism spectrum disorder [59]. Connectome-based fingerprints also show similar relationship: in adolescents, delay in reaching adult-like distinctiveness has been linked to psychiatric symptoms [19] and schizophrenia [22]. On the other hand, periodic activity seems to be a robust individual feature already in childhood, allowing for good differentiation accuracy in children [56]. In our study, we found that differentiability increases with age when fingerprinting in N1 sleep. Decomposing the PSD to periodic and aperiodic components might still refine the age-differentiability relationship in other sleep stages.

With a large children cohort such as the one used in this study, normative modeling [60], akin to growth charts, could be utilized for comparing pediatric patients with developmental conditions to typically developing children. Such approaches have already been successfully applied to functional brain age from sleep EEG [35]. For example, abnormal sleep EEG patterns, such as those present in epilepsy, might manifest in the inter- and intra-subject distances in the latent space projection across and within the sleep stages. Conversely, successful treatment of clinical conditions might manifest as improvement in the differentiability estimates. Building normative models using a latent variable formulation could allow modeling associations between functional neuroimaging data and behavioral scores beyond correlational inferences [61] while accounting for between-subject variance.

Getting representative subject samples with low-density sleep EEG is cheaper and easier compared to other functional neuroimaging modalities, sleep EEG being often the only feasible brain imaging modality to collect data from children. In very young children, awake EEG can have a substantial amount of artifacts that considerably decrease the signal-to-noise ratio and complicates finding meaningful fingerprints. In our analysis, more heterogeneous input data (data from both sleep stages or from segments further apart in the recording) produced more generalizable fingerprints. BRRR –based sleep fingerprints are thus likely to be generalizable to wakefulness as well, but the performance may depend on the states used to train the model. Furthermore, sleep contains oscillatory phenomena that are not present in wakefulness, which may reduce generalizability. Future studies could address this by using data from awake individuals, combined with efficient removal of the more abundant artifacts in the awake EEG.

## Limitations of the study and outlook

The EEG data used in this study was preprocessed only lightly to ensure equal amount of data for all the participants. Therefore, we did not, e.g., remove noisy segments or use automated muscle artifact rejection algorithms. Independent component analysis was applied to remove the heartbeat artifact and any possible eye blinks, and bandpass filtering removed most of the slow drifts, but some muscular and ocular artifacts remained in the data and decreased the differentiability results in N1 sleep. Also, the sleep segments were labeled as "N1" or "N2" sleep based on the emergence of first K-complex alone, but it is likely that the segments contained data from other sleep stages or wakefulness. Furthermore, it is possible that some of the latent components of the BRRR model reflect non-brain activity. We also noticed that the ratio of muscle activity in different sleep stages seemed to be more variable in smaller children, which can make fingerprinting more uncertain in this cohort. Our work was restricted to low-density sensor-level analysis: using source-level features instead might allow investigating which brain areas contribute to individual differentiability and how they might change

across age [11,21], in addition to providing better anatomical accuracy overall. To perform reliable source-level modeling, a higher-density electrode cap of preferably at least 64 electrodes [62] and structural MRIs would be needed. The same holds true for conducting connectivity analyses, which are commonly used in fingerprinting to produce highly individual networks. In addition, differences in EEG cap placement may affect frequency topographies used in fingerprinting. This should be tested in longitudinal data sets with recording sessions conducted in different days.

Even though most frequency-based features used in subject differentiation are derived from oscillatory power in pre-defined frequency bands, such as theta, alpha, beta or gamma bands, considerable individual dynamics of the aperiodic component of PSDs have been reported as well [21,53,63]. EEG fingerprinting based on the aperiodic component of the power spectra, in addition to broadband power, was reported to have high (95%) identification accuracy in the adult population [53]. Developmental and maturation-related trends seem to be reflected in the aperiodic power as well: the oscillatory activity of infants is mostly aperiodic [64], which in the context of our study can mitigate successful across sleep-stage fingerprinting of that cohort.

An important caveat in the interpretation of the PSD-related maturational phenomena relates to the difficulty in determining whether, e.g., the decrease of alpha power during maturation relates to change in alpha amplitude or the incidence frequency of alpha oscillations, especially when using relatively long data segments for estimating PSDs [52], or to physical changes not related to functional maturation. Rapid physical changes during development, such as skull thickening and head size growth, also affect EEG features over time. In future research, source-localized PSDs might allow spatial inferences about the changes in oscillatory features affecting fingerprints. These, however, require MRIs. Head circumference measurements could be used as proxies to scale template MRIs, to inspect to which degree age-related changes in differentiability are attributable to head size. Such an approach was recently used for predicting functional brain age from pediatric sleep EEG, where EEG activity significantly outperformed brain-age prediction based on head circumference [35]. Addressing the oscillatory activity in the temporal domain and in a data-driven manner with, e.g., hidden Markov models [65] could further clarify the individual and age-related dynamics in sleep stages. Another aspect to consider is the possible optimal data time windows for fingerprint estimation, which may vary based on the features used [66]. For resting state, reliable spectral estimates can be achieved from just 30–120 seconds of data [67], but, as demonstrated here, these are subject to changes during the progression of sleep.

## Conclusions

To conclude, we were able to differentiate participants from each other within one recording session with excellent accuracy based on the bandpower estimates of low-density clinical sleep-EEG. The differentiation result held even across sleep stages in a large cohort of children. We found an age effect in the self-differentiability measures of N1 sleep representation, where the differentiability improved as a function of age. Using a probabilistic latent regression model was beneficial for the task, as our model outperformed correlation-based differentiation. Both methods showed similar age relationships in N1 sleep. Within-session sleep fingerprints extracted by latent variable models may help to assess typical development in a normative manner. Further longitudinal research is needed for establishing the presence and evolution of functional brain fingerprints over developmental periods.

## Materials and methods

### Subjects

The dataset contained sleep EEG recordings from 821 healthy Finnish children. The recordings had been originally collected due to clinical concern, but no neurological or developmental diagnoses had been given to the patients within four years after the EEG. The data were measured at Helsinki University Hospital. The age of the children varied between 6 weeks and 19 years (mean ± SD age 4.6 ± 4.3 years) with uneven distribution: over half of the participants were infants (see Fig 7). Of the original data sample, 405 (49.3%) were females, 414 (50.4%) were males and 2 were not assigned

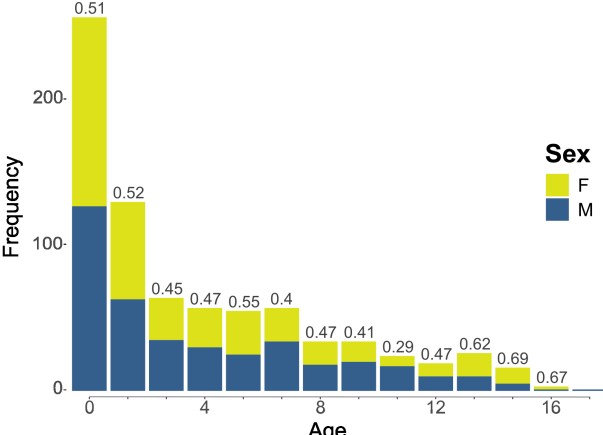

**Fig 7. The age distribution of the subjects.** The proportion of females in each age bin is marked on top of each bar.

sex. The sex ratios were mostly even across the age bins. After removing corrupted or too short data files (N=39), 782 recordings remained for analysis.

### EEG recordings

The EEG sensors were placed using the standard 10-20 montage, and the data was referenced either to the Cz or Pz channel. Eye movements were recorded with piezoelectric sensors. In addition, the recordings contained electromyography (EMG) electrocardiogram (ECG) and breathing sensors. Subjects were measured using two different sizes of caps; generally, children under the age of 5 were measured with a 19-channel cap which lacked frontal (FT) electrodes compared to the one used in older children.

The EEG data segments used in the analysis spanned 900 seconds. The first sleep spindle or K-complex [24], considered to mark the initiation of N2 sleep, was labeled manually by an expert technologist. The preceding 300 seconds were labeled as N1 sleep in the analysis, although from the raw data it was apparent that some of the subjects were periodically awake. The following 600 seconds were categorized as "N2 sleep", although it is possible that the child was in lighter sleep or awake during this stage.

**EEG preprocessing.** Preprocessing of the EEG data was carried out with MNEpython v1.5.1. To have an equal amount of data from each subject, the analysis was limited to EEG channels shared by every recording, totaling 19. All data were re-referenced to an average reference for homogenization.

The EEG time series were bandpass filtered at 1–45 Hz. After filtering, we used independent component analysis (fast-ICA algorithm, [68], implemented in MNE-Python [69]) to project out the prominent ocular and cardiac artifacts. To mimic typical clinical EEG analysis, no additional automatic artifact removal was conducted. All data were manually checked for larger artifacts. Especially during the N2 stage, the data from the sleeping children rarely contained eye blinks, and some of the muscle and movement artifacts were attenuated by the bandpass filtering applied.

Spectral estimates were calculated for each subject after data cleaning. Power spectral densities were calculated separately for both sleep stages (N1 and N2). The EEG time series was cut to 60 second epochs. One-minute data segments are generally considered enough for estimating the spectral properties of rhythmic and arrhythmic phenomena [12,67]. We estimated the power spectral densities (PSDs) for a total of six epochs, 2 for N1 sleep stage and 4 for N2 sleep. More N2 epochs were chosen since the recordings contained twice as much N2 sleep compared to N1 sleep. The PSDs were

calculated using Welch's method with a Hamming window ($n_{fft}$ = 1024, sampling frequency = 250 Hz, frequency resolution ≈ 0.244). Examples of the PSD segments within subjects (in log-scale) are depicted in Fig 1C.

For the regression analysis, 13 separate frequency bands were defined, for which the relative power for each subject was calculated. The width of the frequency bands (1–3 Hz, 3–5.2 Hz, 5.2–7.6 Hz, 7.6–10.2 Hz, 10.2–13 Hz, 13–16 Hz, 16–19.2 Hz, 19.2–22.6 Hz, 22.6–26.2 Hz, 26.2–30 Hz, 30–34 Hz, 34–38.2 Hz, 38.2–42.6 Hz) was set to increase by 0.2 Hz in each consecutive band. To inspect global, age-related changes in the power spectral densities, the subjects were divided into 10 age groups of equal size (refer to the color bar in Fig 1A).

## Modeling

Data analysis was conducted with R v4.4 [70] and MNE python [69]. For an outline of the BRRR analysis process, see Fig 3. We aimed at finding latent spatio-spectral components that would maximally explain the variance between subjects while minimizing the distances between subjects.

The generative BRRR model was built as follows:

$$Y_{n \times s} = (X_{n \times p} \Psi_{p \times k} + \Omega_{n \times k}) \Gamma_{k \times s} + E_{n \times s} \qquad (1)$$

The response matrix $Y_{n \times s}$ contained either two or three PSDs over the 19 channels per subject (here $s$ = 19 channels × 13 frequency bands = 247). Hence, there were 2–3 observations per subject when training the model. The covariate matrix $X_{n \times p}$ was filled with 0–1 -entries, where the observations belonging to the same subject (indicated by the columns) were marked with 1, other entries being zero.

Essentially, $X$ encodes which PSDs belong to which subject. $\Omega_{n \times k}$ carries the unknown latent factors, or components $k$, which are taken to model the latent noise. The low-rank regression coefficient matrix $\Psi_{p \times k}$, together with projection matrix $\Gamma_{k \times s}$, (which projects the latent space to EEG data, or to the observational space) constitute the standard regression coefficient matrix $\Theta = \Psi\Gamma$. Finally, $E_{N \times S}$, $e_i \sim \mathcal{N}(0, \Sigma)$ corresponds to unstructured and independent noise in the observational space. The noise matrix specified in Eq 1 thus contains not only the residual noise, but also the differences within the participant's own data.

The model fit is assessed with the proportion of total variance explained by a rank $K$ BRRR solution, excluding the latent noise. PTVE for the BRRR is defined as in [42]:

$$PTVE = \frac{Tr(\mathrm{Cov}(\hat{Y}))}{Tr(\mathrm{Cov}(Y))} = \frac{Tr(\mathrm{Cov}(X\Psi\Gamma))}{Tr(\mathrm{Cov}(Y))}, \qquad (2)$$

where $Tr$ is short for trace operation.

Conceptually, the BRRR model can be thought of as a combination of factor analysis and regression modeling. In the Bayesian context, the regression coefficients are learned using probabilistic inference. Unlike more customary dimensionality reduction procedures like PCA, BRRR encourages a similar correlation structure to both regression coefficients and target variables via the shared projection $\Gamma$, thus taking into account both the response and covariates in the dimensionality reduction. The product $X\Psi$ constricts the solutions so that the latent factors, or components, are shared when the observations belong to the same subject. For more details of the BRRR model specifications, see [41] and [42].

**Model training and convergence.** Prior to applying the model, the EEG feature matrix $Y$ was Z-scored to have zero mean and unit variance across the features. The latent noise variance prior $\sigma_\Omega^2$ was adjusted to $\frac{1}{10^2}$. Thus, the latent noise levels are assumed to be relatively low, while covariates are allowed to explain most of the variation. The noise priors of the model were tuned to suit the noise levels of EEG data; compared to priors set for MEG data in previous studies [12,13], the expected amount of latent noise was set to be four orders of magnitude larger.

All BRRR models were initialized with Fisher's linear discriminant (LDA) and their parameters were trained using the Gibbs sampler with 1000 iterations. The first 500 samples of the Markov chain Monte Carlo (MCMC) chains were removed as burn-in period, so that the parameter estimates were calculated from the last 500 samples. The convergence of the MCMC chains was assessed by computing the potential scale-reduction factors $\hat{R}$ and bulk- & tail effective sample sizes (ESS) estimated for 200 randomly sampled indices in the regression coefficient matrix $\mathbf{\Theta} = \mathbf{\Psi\Gamma}$, analogously to [41]. The results, indicating good convergence and appropriate sampling of the posterior distribution, are summarized in S2 Fig B: the $\hat{R}$ values are close to 1, and both tail- and bulk-essential sample sizes ($\text{ESS}_b$, $\text{ESS}_t$) are above 100, which is a proposed rules-of-thumb for accepting the sample [71,72].

In addition, model convergence was also reviewed visually by inspecting posterior PTVE traces (S2 Fig A). All checks suggest that 1000 iterations were enough for the Gibbs sampler to reach convergence. For all the convergence checks described here, we used an all-subject model with two segments of N2 data per subject.

**Performance estimation.**

**BRRR.** The input data from the subjects were projected into the $n \times k$ dimensional latent space $\mathbf{Y\Gamma^{-1}}$ inferred by BRRR. To assess how well the latent space structure managed to differentiate between participants, the within-subject distances were compared to between-subject distances and stored in a distance matrix. The minimum L1 distances were then used to determine the model's success rates: the predicted identity of the test subject was the column index where the minimum distance was found. The choice of L1 distance as opposed to Euclidean distance was motivated by its better suitability for high-dimensional data [73].

**Correlation-based fingerprinting.** For bench-marking our BRRR-fingerprinting results, we used a correlation-based fingerprinting on the bandpower estimates. Similarly to previous work on MEG data [11], the distance matrix was calculated using Pearson correlations between observations. The predicted identity was assigned to the column index where the maximum correlation value was found. For this approach, we used all the features in the data without any prior dimensionality reduction. Pearson correlation was opted for as a similarity measure due to its common use in M/EEG and fMRI fingerprinting.

**Effects of feature and observation space sizes.** We studied the BRRR model identity prediction performance as a function of latent space dimension K and number of participants. Here, the predictions were carried out within N2 sleep stage for the whole data set (782 subjects). The results are summarized in Fig 8A. The performance of the BRRR model

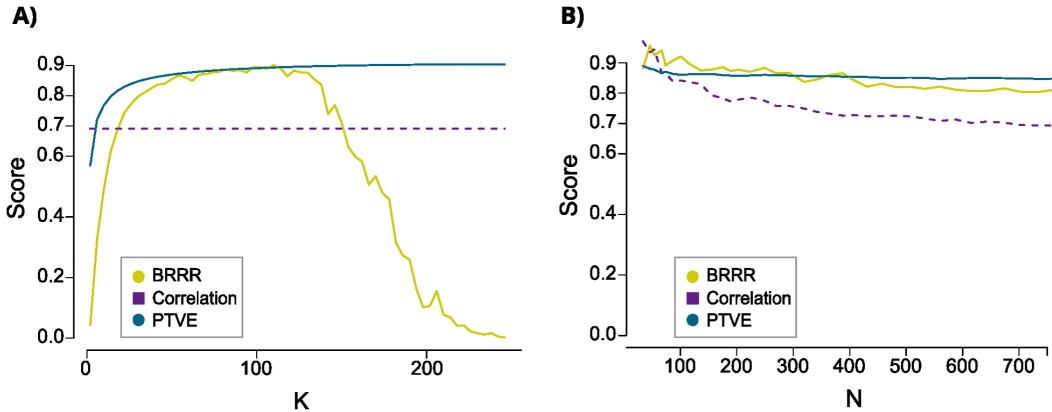

**Fig 8**. **The model performance metrics as a function of latent space dimension and sample size.** Illustrations of prediction success rate and proportion of total variance explained (for BRRR) as a function of latent space dimension K (**A**) and number of subjects N (**B**), while keeping the latent space dimension fixed at $K = 30$. The fingerprinting was carried out with N2 sleep data: here the PTVE and success rate are estimated for the full data set, and both for the BRRR and the correlation-based fingerprinting method. As the correlation-based method uses the full feature space, it does not have K-dependency (**A**).

had an inverse U-like relationship with the latent space dimension: the prediction success rate increased rapidly up to 30 components, reached its maximum of 90% around $K = 100$ and started to decrease rapidly after that, towards near zero at $K = 250$. The PTVE increased rapidly when K increased from 1 to around 50 but improved only marginally after that. The decreasing prediction performance allured to overfitting, as the latent space started to model noise that is not related to the individuality in the data. Between $K = 20 - 150$, BRRR had better prediction performance than the correlation-based method. Based on these investigations, we set the latent space dimension to $K = 30$ using the elbow method. The correlation-based method was more sensitive to the number of randomly sampled participants than the BRRR model was (Fig 8B): the correlation-based method seemed, however, to have an advantage at smaller ($N < 75$) sample sizes. The goodness-of-fit metric for the BRRR model, PTVE, was barely affected by the increasing number of participants after exceeding 100 subjects.

**Cross-validation.** For all subsequent analysis, model prediction accuracy (success rates) and training-PTVEs were estimated by averaging over 10-fold cross-validation results on test data (for success rate) and training data (training-PTVE estimates for model fit). 10-fold cross validation was carried out on the test participant data using corresponding PSD segments that were used for the training subjects. The distances between subjects were computed by projecting the test subjects onto the latent space learned from the training data (Fig 3C).

## Statistical testing

Cluster-level permutation tests [74] were used to determine if the PSD estimates of within N1 or N2 sleep stages differed from each other. The cluster-forming threshold was set to correspond to a p-value of 0.01. The clusters were thresholded using an F-distribution with degrees of freedom $d_1 = 1, d_2 = n_{\text{subj}}$ (N1 sleep) or $d_1 = 3, d_2 = n_{\text{subj}}$ (N2 sleep). The analysis was carried out first for all subjects ($n_{\text{subj}} = 782$), and then for the ten age groups separately ($n_{\text{subj}} = 78$). We also applied cluster-level permutation tests for each age group to assess if the subjects' averages over N1 and N2 segments differed from each other in the sleep spindle band (12–15 Hz). We used the same significance limit as before, and thresholded the clusters using an F-distribution with degrees of freedom $d_1 = 1, d_2 = 78$. To test the overall effect of age on the total power (see Fig 1C and S1 and S2 Tables), here determined as the area under the PSD curve (AUC), we fit a 3rd degree polynomial regression model.

We visualized the latent space projection of the subjects by using t-distributed stochastic neighborhood embeddings [45] (Fig 5), which is a multidimensional scaling method that aims to preserve local distances in the high-dimensional feature space. The differentiability score was defined as the Z-score of within-subject L1-distance relative to others:

$$D_{\text{self}}(i) = -\frac{d(a_i, b_i) - \mu_{d\backslash i}}{\sigma_{d\backslash i}} \qquad (3)$$

where $a_i$ is the first data segment of subject $i$ and $b_i$ is the second data segment of subject $i$, $\mu_{d\backslash i} = \frac{1}{N}\sum_{j\backslash i}^{N} d(a_i, b_j)$ is the average distance to the other subjects and $\sigma_{d\backslash i}$ is the standard deviation of the distance to the other subjects. The negative sign makes this differentiability measure behave similarly to a correlation based Z-score metric [11], where larger scores indicate better discernibility. Mantell's permutation test [75] was employed to assess the Pearson correlation of BRRR and correlation-based dissimilarity matrices. The number of permutations to determine the significance was set to 999.

Furthermore, we used multiple linear regression to examine if cap size (categorical variable), sex or presence of muscle artifacts mediated the correlation-based or BRRR differentiability scores. We summarized the muscle artifacts by calculating the root mean square over the EMG sensors in each sleep segment, followed by applying a logarithm transformation. For example, the regression model explaining differentiability in N2 sleep (using the first two N2 segments) was defined as

$$\hat{D}_{self}(i) = \beta_0 + \beta_1 \cdot noise + \beta_2 \cdot sex + \beta_3 \cdot age + \beta_4 \cdot cap + \beta_5 \cdot (noise \times sex) \tag{4}$$

where $noise = |log(RMS_{N2a}) - log(RMS_{N2b})|$ was the absolute value of the log-ratio of muscle artifacts during the segments used in fingerprinting. For N1 sleep models, we added an interaction term between age and sex. The effect of age on absolute log-ratio on muscle artifact was tested with separate regression models.

## Acknowledgments

We thank the clinical neurophysiology technicians for the recording and marking of the EEG.

### Ethics statement

Ethics statement was obtained from the HUS Regional Committee on Medical Research Ethics (HUS/244/2021), including a waiver of consent due to the retrospective collection of data acquired as part of standard of care.

## Supporting information

**S1 Table. Polynomial regression of N1 area under the curve (AUC) of PSD and age.** Estimates include robust standard errors with 95% confidence intervals.
(PDF)

**S2 Table. Polynomial regression of N2 area under the curve (AUC) of PSD and age.** Estimates include robust standard errors with 95% confidence intervals.
(PDF)

**S3 Table. The 10-fold CV results of different BRRR models.** Models are tabulated for both within and mixed sleep stage models. Fingerprinting task refers to the data used for training and validating the BRRR model with 10% of the population used as validation set. Success rate (SR) and proportion of total variance explained (PTVE) were averaged over the folds. The best results are highlighted.
(PDF)

**S4 Table. Fingerprinting results on whole data.** Results are organized by descending success rate of BRRR fingerprinting. *Corr* refers to correlational fingerprinting. Bolded entries are those that are illustrated in result figures.
(PDF)

**S5 Table. Summary of the generalizability of different BRRR models.** Results are ordered by descending success rates on all data. Results are shown for models trained with all data, with success rates separately for over 7-year old subpopulation. Success rates are shown also for models trained with only >7 year olds. Latent space refers to data segments used in learning the latent representation, whereas fingerprinting task refers to segments projected to latent space that fingerprinting is then performed on.
(PDF)

**S6 Table. Regression results for BRRR and Correlational differentiability scores from N2 sleep.** Here N2 sleep segments are $N2_a + N2_b$. Significant effects are marked with an asterisk: ***)$p < 0.001$, **)$p < 0.01$, *)$p < 0.05$.
(PDF)

**S7 Table. Regression results for BRRR and Correlational differentiability scores from mix of N1 and N2 sleep.** Used $N1_a + N2_b$ data segments. Significant effects are marked with an asterisk: ***)$p < 0.001$, **)$p < 0.01$, *)$p < 0.05$.
(PDF)

**S1 Fig. Subjects in N2 latent space projection.** Two-dimensional t-SNE projection of the subjects projected on 30-dimensional latent mapping provided by BRRR (trained with N2 data). The first N2 segment is marked with a circle, the second with a triangle. As in Fig 5, we observe an age effect but not sex effect in the latent space.
(TIFF)

**S2 Fig. Multiple regression of correlational differentiability scores in N1 sleep**. The differentiability and artifact-sex interaction effect is illustrated on the left (**A**: $\beta_{\text{Female}} = -0.16, \beta_{\text{Male}} = -0.99$), **B**: the marginal age effect on differentiability ($\beta_{\text{Female}} = 0.05, \beta_{\text{Male}} = 0.02$).
(TIFF)

**S3 Fig. The age dependency of log-muscle artifact ratio.** Left figure contains results for N1 sleep ($\beta = -0.0076, p < 0.001, r^2 = 0.045$), right for mix of N1 and N2 sleep ($\beta = -0.0098, p < 0.001, r^2 = 0.054$).
(TIFF)

**S4 Fig. Model convergence diagnostics.** Fig **A** depicts a trace plot of the model PTVE across iterations (after discarding the first half of the samples as burn-in period) along with Markov chain convergence diagnostics, demonstrating good convergence. Table **B** summarizes the converge diagnostics (mean $\pm$ SD) of 200 randomly sampled indices from the low-rank regression coefficient matrix.
(TIFF)

## Author contributions

**Conceptualization:** Verna Heikkinen, Susanne Merz, Riitta Salmelin, Mia Liljeström, Hanna Renvall.

**Data curation:** Verna Heikkinen, Sampsa Vanhatalo, Leena Lauronen.

**Formal analysis:** Verna Heikkinen.

**Funding acquisition:** Hanna Renvall.

**Investigation:** Verna Heikkinen, Susanne Merz, Mia Liljeström, Hanna Renvall.

**Project administration:** Hanna Renvall.

**Resources:** Sampsa Vanhatalo, Leena Lauronen.

**Software:** Verna Heikkinen, Susanne Merz.

**Supervision:** Mia Liljeström, Hanna Renvall.

**Visualization:** Verna Heikkinen.

**Writing – original draft:** Verna Heikkinen, Mia Liljeström, Hanna Renvall.

**Writing – review & editing:** Verna Heikkinen, Susanne Merz, Riitta Salmelin, Sampsa Vanhatalo, Leena Lauronen, Mia Liljeström, Hanna Renvall.

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
