## [Decision Letter · Decision Letter 0]

24 Sep 2025

PCOMPBIOL-D-25-01108

Latent-noise regression model captures individual variation in children’s oscillatory brain activity during nREM sleep electroencephalogram

PLOS Computational Biology

Dear Dr. Heikkinen,

Thank you for submitting your manuscript to PLOS Computational Biology. After careful consideration, we feel that it has merit but does not fully meet PLOS Computational Biology's publication criteria as it currently stands. Therefore, we invite you to submit a revised version of the manuscript that addresses the points raised during the review process.

Please submit your revised manuscript within 60 days Nov 24 2025 11:59PM. If you will need more time than this to complete your revisions, please reply to this message or contact the journal office at ploscompbiol@plos.org. Please include the following items when submitting your revised manuscript:

We look forward to receiving your revised manuscript.

Kind regards,

Christian Keitel

Academic Editor

PLOS Computational Biology

Hugues Berry

Section Editor

PLOS Computational Biology

**Editor Comments :**

Two expert reviewers made the following comments and suggestions that, I agree, would need to be addressed in a revised version of the submission: Reviewer #1 raises some points over contextualising the presented approach within the larger brain-fingerprinting literature, with a focus on what the specific advantage of the taken approach is. Reviewer #2 requires some clarification regarding the EEG data quality and treatment (or non-treatment) of artifacts.

**Journal Requirements:**

3) We notice that your supplementary Figures, and Tables are included in the manuscript file. Please remove them and upload them with the file type 'Supporting Information'. Please ensure that each Supporting Information file has a legend listed in the manuscript after the references list.

4) Please amend your detailed Financial Disclosure statement. This is published with the article. It must therefore be completed in full sentences and contain the exact wording you wish to be published.

2) If any authors received a salary from any of your funders, please state which authors and which funders.

**Reviewers' comments:**

Reviewer's Responses to Questions

Reviewer #1: Heikkinen et al. explore whether children and young adults are both accurately differentiated from brain-fingerprints derived from resting-state EEG across various sleep stages. The methods and data reported are very unique and offer a complementary perspective to the growing brain-fingerprinting literature. The paper is interesting and relevant to the community. Below, I have detailed several points that I believe will enhance the clarity of the manuscript and better highlight the specific contributions of these findings to the broader brain-fingerprinting literature.

Major

1. The literature review lacks any reference to previous developmental brain-fingerprinting studies. There has been considerable work examining how the stability of functional connectomes throughout development may serve as a biomarker for health, specifically mental health. See for example (Fu et al., 2023; Kaufmann et al., 2017, 2018).

I believe that the paper would benefit from contextualizing the current findings with respect to the broader brain-fingerprinting literature. This will help frame the paper and explain its unique contribution to the literature.

2. The authors note in the introduction that: “Previous approaches to find individually stable features from functional neuroimaging data have generally relied on correlations between different measurements of the same participants. While correlation approaches can reliably identify individuals within specific subject groups, such as patients and controls (17), it has remained uncertain which factors contribute to the differentiability of any individual subject.”

It is unclear to me whether they are referring to salient features for brain-fingerprinting or to specific demographic factors that may alter the brain-fingerprint of individuals. Could the authors clarify this point? In either case, I would argue that the correlational brain-fingerprint methodology can capture both features using group-level statistics. Overall, I am unclear about the specific advantage of the BRRR approach. Both approaches can derive the specific features which drive participant differentiation, albeit in different ways.

Take, for example, their age-self-similarity effect in the latent space. Could the authors show that this effect is best captured by the BRRR approach over correlational brain-fingerprinting methods?

3. How do you interpret the differences in stability of the spectra between sleep stages (Figure 2) concerning neurodevelopmental effects? The point the authors are trying to make is unclear to me. From the introduction, I would have thought the authors would explore an age-by-sleep-stage interaction on the stability of PSD estimates. Yet, they report that most of the age groups show differences in the stability of spectra across sleep stages. Therefore, I am unclear how the authors interpret the findings of Figure 2.

I interpret these findings to suggest that high-frequency bands are unstable between sleep stages for all age groups.

On a related note, the authors should specify how they dealt with regional differences in the power spectra. Did they fit this model for the average spectra across all electrodes?

4. The methodology of the ‘null models’ is unclear to me. Based on the correlational brain-fingerprinting method, no model is trained. The similarity between two data segments/ recordings are analyzed; therefore, it is unclear how the model is validated on an ‘out-of-sample’. Moreover, it is unclear how one can fingerprint with more than one recording/ segment. Overall, the methodology of Figure 4 is very unclear to me. I believe the paper and the interpretability of the results would be greatly improved if the authors clarified this point.

5. How does the self-similarity effect compare to traditional brain-fingerprinting measures (i.e., self-similarity as measured by correlation)? I believe that the inclusion of these analyses would make the results easier to interpret.

I also want to note that the decreasing self-similarity with age is the opposite prediction of the brain-fingerprint literature (Fu et al., 2023; Kaufmann et al., 2017, 2018). Similarly, it seemingly contradicts the authors’ own predictions, “model trained using older participants could provide more stable fingerprint” and “sleep patterns would become increasingly stable with age”. These findings therefore, are very difficult to interpret without a benchmark from more traditional self-similarity measures. Is the model introducing an age bias, or is it something specific to sleep stages or the dataset itself? Could the authors explicitly state why they believe the results are in the opposite direction of their hypothesis and the literature.

6. The results section could be reorganized. The panels of each of the Figures are not discussed in order, which makes it difficult for the reader to follow the figures. For example, the section “BRRR fingerprints generalize better compared to correlation-based fingerprinting” which is about Figure 4, is presented after Figure 6.

7. The authors note: “Thus, a similarity metric based on just correlation does not appear to generalize across sleep stages and suffers from increased dimensionality and heterogeneity of the training data.”

One could argue that this lack of stability is very interesting and something to be understood and not reduced with the BRRR model. For example, previous literature reports that the brain-fingerprints of individuals with Parkinson’s disease are less stable than age-matched controls (da Silva Castanheira, Wiesman, Hansen, et al., 2024; Troisi Lopez et al., 2023). This is specifically driven by arrhythmic brain activity. I believe that exploring the instability of brain activity is a useful approach to understanding why participants are less differentiable from the brain activity and may offer novel biological insights. Indeed, the authors themselves highlight this in the introduction: “The stability of these individual features, or the lack thereof, could similarly serve as a surrogate biomarker of risk of atypical development.”

I am not saying that there is no added value of the BRRR approach, but I believe that the authors can do a better job at contextualizing how and when the BRRR approach is useful while also highlighting some advantages of the correlational brain-fingerprinting methods approach.

8. More generally, I feel like a discussion on the utility of a BRRR brain-fingerprint model that can predict inter-individual differences across many physiological states would benefit the paper’s structure. Sleep and wakefulness represent very different mental and physiological states, and as such, the functional connectivity of the brain and its dynamics are radically different. Should we expect to be able to differentiate individuals across these states based on functional activity? Could one not argue that the salient features for participant differentiation across various physiological states are driven by anatomical and other non-electrophysiological factors, which are unlikely to be altered by physiological states?

9. In line with the point above, what steps did the authors take to address non-electrophysiological confounds?

Minor

1. The authors discuss how they believe sleep EEG is a great opportunity to study children, because of the inherent difficulty in collecting neural data from this population. I would appreciate it if the authors could discuss specific challenges of generalizing sleep EEG-derived fingerprints to wakefulness versus artifact correction on brain-fingerprints derived from awake individuals.

2. How did the authors test the non-linearity of the spectral AUC effect? To me, it seems like they ran a series of t-tests, but I believe a polynomial regression model would be a better test of the non-linear effect.

On a more general note, I believe that including a sentence or two before each result explaining the methodology would be helpful to the reader. It would provide them with important context on which statistical test was performed.

3. How do you ensure the placement of electrodes is comparable across age groups, given the very different head sizes with age?

4. I find the red lines in Figure 4 hard to read. I would recommend that the authors find a different way of depicting their results.

5. The authors “hypothesized that the model trained using older participants could provide more stable fingerprints compared to the model using all participants, especially across sleep stages.” Yet, it seems like spectral features are less stable for the older age group (Figure 2), i.e., more grey segments with age, especially for the N2. How do the authors reconcile these effects?

6. For consistency, I would flip the colour bar of Figure 3d such that the colours in both matrices correspond to the same thing (i.e., hot colours mean more similar).

7. The name self-distance is misleading in Figure 4c and the subsequent analyses. The authors define a z-score-like metric of distance in the latent space, not the absolute distance between a participant’s two recordings. Therefore, when looking at the difference between self-similarity and between-subjects similarity, we observe large differences, but this comparison is not fair as the metrics are on different scales (one being an absolute measure of distance and the other relative). I would recommend that the authors compute the ‘raw’ self-similarity distance metric for their analyses or rename this metric to something akin to differentiability (which I believe it more closely measures).

8. The authors speculate that the effects observed in the low-dimensional space might be driven by arrhythmic features, but this is precisely what traditional brain fingerprinting approaches would be able to tell you (da Silva Castanheira, Wiesman, Hansen, et al., 2024; da Silva Castanheira, Wiesman, Taylor, et al., 2024).

How does the BRRR model improve interpretability? I think the authors convincingly show that the BRRR method improves accuracy, but they do little to examine how it impacts the salient features for participant differentiation (which one may argue is easier to do on brain-fingerprints not defined in a latent space). Again, I want to emphasize that I believe that the BRRR method has specific advantages, but these are not immediately evident when reading the paper and can be made more obvious. Perhaps a thorough discussion of the advantages of each technique and their potential application would help clarify this point.

References

da Silva Castanheira, J., Wiesman, A. I., Hansen, J. Y., Misic, B., Baillet, S., Breitner, J., Poirier, J., Bellec, P., Bohbot, V., & Chakravarty, M. (2024). The neurophysiological brain-fingerprint of Parkinson’s disease. EBioMedicine, 105. https://www.thelancet.com/journals/ebiom/article/PIIS2352-3964(24)00236-6/fulltext

da Silva Castanheira, J., Wiesman, A. I., Taylor, M. J., & Baillet, S. (2024). The Lifespan Evolution of Individualized Neurophysiological Traits. bioRxiv: The Preprint Server for Biology, 2024.11.27.624077. https://doi.org/10.1101/2024.11.27.624077

Fu, Z., Liu, J., Salman, M. S., Sui, J., & Calhoun, V. D. (2023). Functional connectivity uniqueness and variability? Linkages with cognitive and psychiatric problems in children. Nature Mental Health, 1(12), Article 12. https://doi.org/10.1038/s44220-023-00151-8

Kaufmann, T., Alnæs, D., Brandt, C. L., Bettella, F., Djurovic, S., Andreassen, O. A., & Westlye, L. T. (2018). Stability of the Brain Functional Connectome Fingerprint in Individuals With Schizophrenia. JAMA Psychiatry, 75(7), 749–751. https://doi.org/10.1001/jamapsychiatry.2018.0844

Kaufmann, T., Alnæs, D., Doan, N. T., Brandt, C. L., Andreassen, O. A., & Westlye, L. T. (2017). Delayed stabilization and individualization in connectome development are related to psychiatric disorders. Nature Neuroscience, 20(4), Article 4. https://doi.org/10.1038/nn.4511

Troisi Lopez, E., Minino, R., Liparoti, M., Polverino, A., Romano, A., De Micco, R., Lucidi, F., Tessitore, A., Amico, E., Sorrentino, G., Jirsa, V., & Sorrentino, P. (2023). Fading of brain network fingerprint in Parkinson’s disease predicts motor clinical impairment. Human Brain Mapping, 44(3), 1239–1250. https://doi.org/10.1002/hbm.26156

Reviewer #2: The study presented in this work explores the possibility of neural fingerprinting in a pediatric population. Using a probabilistic modeling approach on a large retrospective sleep-EEG dataset, the authors demonstrate that individual neural fingerprints derived from EEG bandpower are present from a young age, and suggest an increasing stability with age.

However, I have a number of concerns regarding some methodological aspects of this study:

BRRR MODEL DEFINITION

The dimensionality of the latent space is fixed to k = 30. The definition of this parameter appears arbitrary. How was this value chosen?

EFFECT OF AGE ON FINGERPRINTING ACCURACY

A substantial part of this work’s claims relies on the increasing stability of individual neural fingerprints with age. However, these results stem from the comparison of a model fitted to the entire cohort with a second model fitted only participants older than 7. First, the definition of the older group appears subjective, particularly as it does not match the age groups used in the first analyses. What is the rationale supporting a clear split from 7 years onwards? Next, comparing the performance of the model trained with all ages to one trained with only participants closer in age might be biased, as more subtle features could become of importance in the latter case. How does the accuracy compare for the >7 y.o. between both models?

To provide a stronger support to this hypothesis, it would be of interest to show the recognition accuracy by age group, similarly to Figure 4A/B.

DATA QUALITY AND EEG PREPROCESSING

A number of sentences throughout the text raise some concerns about the quality of the data and a possible confounding role of movement artefacts.

Lines 519-520: “although from the raw data it was apparent that some of the subjects were periodically awake “.

Lines 532-534: “All the data was manually checked for larger artifacts. Especially during the N2 stage, the data from the sleeping children rarely contained any blinks, and some of the muscle and movement artifacts were attenuated by the bandpass filtering applied”.

Lines 361-362: “Possible differences between sexes may also be attributable to

external factors, such as non-equal distribution of movement artifacts”

Lines 433-438: “we did not, e.g., remove noisy segments or use automated artifact rejection algorithms. Independent component analysis was applied to remove the heartbeat artefact, and bandpass filtering took out most of the slow drifts, but some muscular and ocular artefacts likely remained in the data. Any discrepancy in artefact density, such as uneven amount of movement between sexes in N1 sleep, might account for some of the observed sex differences.”

How were these artefacts and apparent periods of wakefulness distributed across subjects and ages? In particular, were the concerned subjects displaying higher accuracy (suggesting these non-brain signals contributed strongly to their “fingerprint”) or lower (thus artificially reducing the decoding power of the model)? An examination of the latent components might help answering these questions.

CHANGES IN SPECTRAL POWER DURING CHILDHOOD

Lines 116-117 mention a pattern specific to infants <3 months. No results are displayed for this age group, and this grouping is not coherent with the age groups used for the other analyzes. Which information supports this claim ?

LANGUAGE AND WRITING

Line 19: fMRI has not been defined

Line 268: (p¿0.05)

**Have the authors made all data and (if applicable) computational code underlying the findings in their manuscript fully available?**

Reviewer #1: **No:** Data not shared. Did not check the code.

Reviewer #2: None

PLOS authors have the option to publish the peer review history of their article (what does this mean?). If published, this will include your full peer review and any attached files.

Reviewer #1: **Yes:** Jason da Silva Castanheira

Reviewer #2: No

**Figure resubmission:**
---

## [Decision Letter · Decision Letter 1]

20 Jan 2026

Dear Doctoral researcher Heikkinen,

We are pleased to inform you that your manuscript 'Capturing individual variation in children’s electroencephalograms during nREM sleep' has been provisionally accepted for publication in PLOS Computational Biology.

Best regards,

Hugues Berry

Section Editor

PLOS Computational Biology

Hugues Berry

Section Editor

PLOS Computational Biology

Reviewer's Responses to Questions

**Comments to the Authors:**

Reviewer #1: The authors have addressed all of my concerns thoroughly and thoughtfully.

Reviewer #2: I thank the authors for their revision and responses to the aspects I have raised. I believe the manuscript improved substantially.

**Have the authors made all data and (if applicable) computational code underlying the findings in their manuscript fully available?**

Reviewer #1: None

Reviewer #2: None

PLOS authors have the option to publish the peer review history of their article (what does this mean?). If published, this will include your full peer review and any attached files.

Reviewer #1: **Yes:** Jason da Silva Castanheira

Reviewer #2: No

---

## [Editor Report · Acceptance letter]

PCOMPBIOL-D-25-01108R1

Capturing individual variation in children’s electroencephalograms during nREM sleep

Dear Dr Heikkinen,

I am pleased to inform you that your manuscript has been formally accepted for publication in PLOS Computational Biology. Your manuscript is now with our production department and you will be notified of the publication date in due course.

With kind regards,

Anita Estes
